# Freelance Holography, Part II:
## Moving Boundary in Gauge/Gravity Correspondence

**A. Parvizi**[a] , **M.M. Sheikh-Jabbari**[b] , **V. Taghiloo**[b,c]

[a] *University of Wroclaw, Faculty of Physics and Astronomy, Institute of Theoretical Physics,*
*Maksa Borna 9, PL-50-204 Wroclaw, Poland*

[b] *School of Physics, Institute for Research in Fundamental Sciences (IPM),*
*P.O.Box 19395-5531, Tehran, Iran*

[c] *Department of Physics, Institute for Advanced Studies in Basic Sciences (IASBS),*
*P.O. Box 45137-66731, Zanjan, Iran*

*E-mail:* aliasghar.parvizi@uwr.edu.pl, jabbari@theory.ipm.ac.ir,
v.taghiloo@iasbs.ac.ir

ABSTRACT: We continue developing the freelance holography program, formulating gauge/gravity correspondence where the gravity side is formulated on a space bounded by a generic timelike codimension-one surface inside AdS and arbitrary boundary conditions are imposed on the gravity fields on the surface. Our analysis is performed within the Covariant Phase Space Formalism (CPSF). We discuss how a given boundary condition on the bulk fields on a generic boundary evolves as we move the boundary to another boundary inside AdS and work out how this evolution is encoded in deformations of the holographic boundary theory. Our analyses here extend the extensively studied $T\bar{T}$-deformation by relaxing the boundary conditions at asymptotic AdS or at the cutoff surface to be any arbitrary one (besides Dirichlet). We discuss some of the implications of our general freelance holography setting.

# Contents

---

# 1  Introduction

AdS/CFT [1–3] is the best formulated example of holography [4–8] that establishes a duality between a quantum gravitational bulk theory in an asymptotically AdS spacetime and a quantum field theory (QFT) defined on its asymptotic timelike boundary. In its standard canonical form, this duality is defined by imposing Dirichlet boundary conditions on the bulk fields at the AdS boundary. In the large-N limit, where the bulk side reduces to a classical gravity theory, the duality reduces to the gauge/gravity correspondence. In this limit, the possibility of having arbitrary boundary conditions on bulk fields has been explored. It is well known that modifying bulk boundary conditions corresponds to introducing multi-trace deformations in the boundary theory [9–11] (and for a systematic treatment of multi-trace deformations see also [12–15] and also [16–33]). In [34], we presented a systematic analysis of formulating gauge/gravity correspondence beyond the Dirichlet boundary conditions within the covariant phase space formalism (CPSF) framework [35–38] (for review see [39–41]).

CPSF provides a systematic framework for studying the solution phase space of any theory with any consistent set of boundary conditions. In a generic field theory, the solution phase space possesses a symplectic form, typically composed of codimension-one and codimension-two integrals. The former involves bulk modes, while the latter involves boundary modes, which are physical degrees of freedom residing exclusively at the boundary and lacking bulk propagation. Importantly, the existence of boundary modes is independent of the precise boundary conditions applied to bulk fields. This raises the question: what determines the dynamics of these boundary modes? The answer is found in the interplay with the bulk field's boundary conditions. Specifically, boundary modes dynamically adjust to ensure the bulk fields adhere to the imposed constraints. As a result, modifying the boundary conditions of the bulk fields necessarily alters the dynamics of the boundary modes to maintain consistency. This interplay—where boundary modes reorganize in response to changing bulk boundary conditions—provides a physical interpretation of boundary multi-trace deformations.

In this paper, we explore whether holography remains valid at a finite radial cutoff in asymptotically AdS spacetimes. This question has been studied in the context of AdS/CFT, particularly in lower dimensions. In three-dimensional ($3d$) gravity, it has been formulated, with the answer being affirmative. As shown in [42], achieving a holographic dual at a finite cutoff with Dirichlet boundary conditions requires deforming the boundary theory by the $T\bar{T}$ operator [43, 44]. This deformation, constructed from a specific quadratic combination of the boundary energy-momentum tensor, exhibits remarkable properties, including integrability [44, 45]. It also allows for exact computations of various quantities, such as the S-matrix [46–48], the deformed Lagrangian [45, 49, 50], and the finite-volume energy spectrum [44, 45]. Moreover, it preserves key features of the original theory, including modular invariance and the Cardy formula for

entropy [51–53]. These properties ultimately stem from Zamolodchikov's factorization formula for the $T\bar{T}$ operator [43] (for a nice review, see [54]).

The question of holography at a finite cutoff in higher dimensions was explored in [55, 56], where a higher-dimensional generalization of the $T\bar{T}$ deformation was introduced. However, unlike its two-dimensional ($2d$) counterpart, the higher-dimensional version lacks the factorization property, which only emerges in the large-N limit.

In this paper, we use the covariant phase space formalism (CPSF) to develop further the $T\bar{T}$ deformation in arbitrary dimensions and to cases where the boundary conditions at finite cutoff can be chosen arbitrarily. Our approach illustrates how one can start from the standard gauge/gravity correspondence at asymptotic infinity and, through the tools of CPSF, systematically construct a holographic dual at a finite radial cutoff. Previous constructions of finite-cutoff holography have been limited to Dirichlet boundary conditions at the cutoff surface [42, 55, 56]. Here, we extend the framework by allowing for arbitrary boundary conditions, thereby broadening the scope of the duality. This extension constitutes the core of *Freelance Holography II*, which seeks to establish a more general class of finite-distance holographic correspondences.

Gravity with a finite radial cutoff and *Dirichlet* boundary conditions is generally ill-defined [57–64]. Thus, constructing a well-defined holographic dual at a finite distance necessitates modifying the boundary conditions of the bulk fields. In this work, we provide a systematic framework for implementing such modifications, extending holography beyond the standard Dirichlet setup. In particular, the $T\bar{T}$ deformation allows us to map the conventional gauge/gravity correspondence—formulated with Dirichlet boundary conditions at the asymptotic boundary—onto a holographic description at a finite cutoff with Dirichlet conditions imposed at the cutoff surface [42, 55, 56]. In this work, we introduce a broader class of deformations that enable transitions between holographic formulations with arbitrary boundary conditions at different radii. As specific cases, we analyze setups with Dirichlet, Neumann, and conformal boundary conditions.

In the literature, the $T\bar{T}$ deformation has been interpreted in two distinct ways [42, 65]. The first interpretation views it as a radial flow, where the AdS boundary moves inward, leading to a holographic description with Dirichlet boundary conditions imposed at a finite cutoff surface [42]. The second interpretation, in contrast, does not involve a radial evolution but instead treats the $T\bar{T}$ deformation as a modification of the bulk boundary conditions at the asymptotic AdS boundary [65]. In this paper, we establish the equivalence of these two perspectives for our extended $T\bar{T}$ framework, holding for arbitrary boundary conditions on the bulk fields in arbitrary dimensions.

The main idea behind the Freelance Holography program is to employ freedoms (also called ambiguities) in the CPSF to set the boundary and boundary conditions free in the holographic framework. In this paper, we leverage finite-cutoff holography to fix both bulk and boundary freedoms/ambiguities systematically using the holographic duality. In particular, we demonstrate that the bulk $Y$-freedom is uniquely determined in terms of the boundary symplectic potential, providing a more refined and physically motivated formulation of the CPSF.

Finally, we introduce two classes of hydrodynamic deformations that map one hydrodynamic system to another. The first class, extrinsic hydrodynamic deformations, encompasses deformations associated with Dirichlet, Neumann, and conformal boundary conditions as special cases. In $d = 2$, we also show that the radial evolving deformations ($T\bar{T}$) admit hydrodynamical description.

**Outline of the Paper.** Section 2 contains reviews of basic formulations we use: we introduce the geometric setup, briefly review the covariant phase space formalism (CPSF), the gauge/gravity correspondence, and Freelance Holography I [34]. In Section 3, starting from the standard gauge/gravity correspondence, we construct holography at a finite cutoff with Dirichlet boundary conditions and derive the corresponding boundary theory as a deformation of the asymptotic boundary theory. In Section 4, we extend finite-cutoff holography to accommodate arbitrary boundary conditions. In Section 5, we explore interpolation between different boundary conditions at two distinct radial locations. A special case involves interpolating between the asymptotic AdS boundary and an interior cutoff surface. In Section 6, we examine examples from general relativity with various matter fields in different spacetime dimensions and derive explicit forms of radial evolution deformations. In Section 7, we analyze transitions between different boundaries with varying boundary conditions in pure general relativity across multiple dimensions. In Section 8, we introduce two classes of hydrodynamic deformations and examine the $T\bar{T}$ deformation in this context. In Section 9, we summarize our findings, discuss various aspects of Freelance Holography, and outline potential future directions. In Appendices A and B, we provide technical details and computations.

## 2 Review and the setup

In this section, we establish the foundational framework for our discussion by reviewing key concepts and setting up the geometric and formal structure necessary for our analysis. We begin with a description of the geometric setup, where we define a foliation of asymptotically AdS spacetime by a family of timelike hypersurfaces. Next, we provide a concise review of the Covariant Phase Space Formalism (CPSF), which serves as a crucial tool for analyzing the space of solutions with arbitrary boundary conditions. Following this, we briefly revisit the fundamental principles of the gauge/gravity correspondence, outlining the standard AdS/CFT dictionary and its large-N limit, where the duality simplifies to a classical gauge/gravity correspondence. This prepares us for the modifications introduced by our "Freelance Holography" program. Finally, we summarize the key results from "Freelance Holography I" [34], where we extended the gauge/gravity duality beyond the usual Dirichlet boundary conditions.

### 2.1 Geometric setup

Consider a $(d+1)$-dimensional asymptotically AdS spacetime $\mathcal{M}$ with coordinates $x^\mu$ and metric $g_{\mu\nu}$. We introduce a foliation of $\mathcal{M}$ by a family of codimension-one timelike hypersurfaces, denoted as $\Sigma_r$. For brevity, we refer to the asymptotic timelike boundary of AdS, $\Sigma_{r=\infty}$, simply as $\Sigma$.

To describe this setup, we decompose the spacetime coordinates as $x^\mu = (x^a, r)$, where $a = 0, 1, \ldots, d-1$. Here, $x^a$ are intrinsic coordinates on each hypersurface $\Sigma_r$, while $r \in [0, \infty]$ serves as a radial parameter labeling the hypersurfaces, with $r = \infty$ representing the AdS causal boundary. This formulation allows us to interpret $r$ as a radial coordinate, with $\Sigma_r$ corresponding to constant-$r$ timelike slices of spacetime.

We define $\mathcal{M}_c$ as the portion of AdS spacetime with $r \leq r_c$ and denote its boundary as $\Sigma_c$ (a shorthand for $\Sigma_{r_c}$). Similarly, $\mathcal{M}_r$ refers to the region of asymptotically AdS space bounded by $r$, with $\Sigma_r$ as its corresponding boundary.

Using this foliation, one can naturally perform a radial $(1 + d)$-dimensional ADM decomposition—a generalized Fefferman-Graham expansion—of the line element, which takes the standard form

$$\mathrm{d}s^2 = N^2 \, \mathrm{d}r^2 + h_{ab}(\mathrm{d}x^a + U^a \, \mathrm{d}r)(\mathrm{d}x^b + U^b \, \mathrm{d}r) \,, \tag{2.1}$$

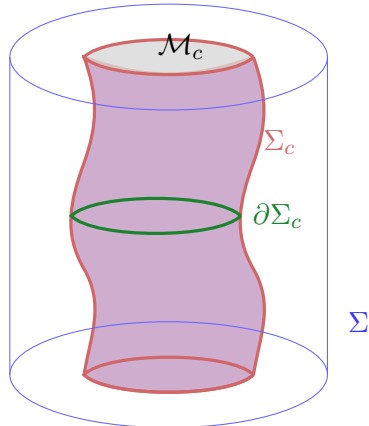

**Figure 1**: Portion of an asymptotically AdS spacetime bounded by $r \leq r_c$ with a generic timelike boundary $\Sigma_c$. We formulate physics in the shaded region $\mathcal{M}_c$.

where $N$ is the radial lapse function, $U^a$ is the radial shift vector, and $h_{ab}$ is the induced metric on the codimension-one constant-$r$ timelike hypersurface $\Sigma_r$. For later convenience, we introduce the conformally rescaled induced metric as $h_{ab} = r^2 \gamma_{ab}$. [1] The one-form normal to $\Sigma_r$ and the associated induced metric are given by

$$s = s_\mu \, \mathrm{d}x^\mu = N \, \mathrm{d}r \,, \qquad h_{\mu\nu} = g_{\mu\nu} - s_\mu s_\nu \,. \tag{2.3}$$

For later use, we introduce the extrinsic curvature of $\Sigma_r$, given by

$$K_{\alpha\beta} = \frac{1}{2} h^\mu_\alpha \, h^\nu_\beta \, \mathcal{L}_s h_{\mu\nu} \,, \qquad \text{with} \qquad h^\mu_\alpha = \delta^\mu_\alpha - s^\mu s_\alpha \,. \tag{2.4}$$

Its explicit form is

$$K_{ab} = \frac{1}{2N} \mathcal{D}_r h_{ab} \,, \qquad \mathcal{D}_r := \partial_r - \mathcal{L}_U \,, \tag{2.5}$$

where $\mathcal{L}_U$ denotes the Lie derivative along the shift vector $U^a$. For convenience, we introduce the following shorthand notation for integrals:

$$\int_\mathcal{M} := \int_\mathcal{M} \mathrm{d}^{d+1}x \,, \qquad \int_{\Sigma_r} := \int_{\Sigma_r} \mathrm{d}^d x \,, \qquad \int_{\partial\Sigma_r} := \int_{\partial\Sigma_r} \mathrm{d}^{d-1}x \,. \tag{2.6}$$

With this notation, Stokes' theorem takes the form

$$\int_{\mathcal{M}_r} \sqrt{-g} \, \nabla_\mu X^\mu = \int_{\Sigma_r} \sqrt{-g} \, X^r = \int_{\Sigma_r} \sqrt{-h} \, s_\mu X^\mu \,. \tag{2.7}$$

Here, $X^\mu$ represents an arbitrary vector field in spacetime which we assume to be smooth over $\mathcal{M}_r$, and we have utilized the determinant relation $\sqrt{-g} = N\sqrt{-h}$.

## 2.2 Brief review of covariant phase space formalism

In this subsection, we provide a brief review of the covariant phase space formalism, which is presented in terms of $(p, q)$-forms: $p$-forms over the spacetime and $q$-forms over the solution space. We use boldface

---

[1]The metric of the asymptotically AdS spacetime in the standard Fefferman-Graham coordinates is given by:

$$ds^2 = \ell^2 \frac{dr^2}{r^2} + r^2 \gamma^0_{ab} dx^a dx^b + \cdots \,, \tag{2.2}$$

where $\ell$ is the AdS radius, and the ellipsis represents higher-order terms in the expansion.

symbols to denote these $(p, q)$-forms. The Lagrangian of any $D$-dimensional theory, denoted by $\mathbf{L}$, is a top-form, specifically a $(D, 0)$-form. Thus, we have

$$\mathrm{d}\mathbf{L} = 0\,, \qquad \delta\mathbf{L} \overset{\circ}{=} \mathrm{d}\boldsymbol{\Theta}\,, \tag{2.8}$$

where $\overset{\circ}{=}$ denotes on-shell equality, and d and $\delta$ represent exterior derivatives over spacetime and solution space, respectively. The quantity $\boldsymbol{\Theta}$ is the symplectic potential, a $(D-1, 1)$-form. Eq.(2.8) may be viewed as the equation defining $\boldsymbol{\Theta}$. It may be promoted to a definition in which the Lagrangian of the theory does not explicitly appear

$$\mathrm{d}\delta\boldsymbol{\Theta} \overset{\circ}{=} 0\,. \tag{2.9}$$

That is, by definition, $\boldsymbol{\Theta}$ is a $(D-1, 1)$-form that is closed both on spacetime and on solution space. The above can be "integrated" over the spacetime and the solution space, yielding a solution with "integration constants":

$$\boldsymbol{\Theta} = \boldsymbol{\Theta}_{\mathrm{D}} + \delta\mathbf{W} + \mathrm{d}\mathbf{Y} + \mathrm{d}\delta\mathbf{Z}\,, \tag{2.10}$$

where $\boldsymbol{\Theta}_{\mathrm{D}}$ is the specific solution satisfying $\mathrm{d}\delta\boldsymbol{\Theta}_{\mathrm{D}} = \delta\,\mathrm{d}\boldsymbol{\Theta}_{\mathrm{D}} = 0$, chosen to enforce the Dirichlet boundary condition for the fields of the theory. Additionally, when the boundary on which this symplectic potential is evaluated approaches the asymptotic boundary, we require $\boldsymbol{\Theta}_{\mathrm{D}}$ to remain finite.[2] The fields $\mathbf{W}$, $\mathbf{Y}$, and $\mathbf{Z}$ are $(D-1, 0)$, $(D-2, 1)$, and $(D-2, 0)$-forms, respectively, representing integration constants and the three degrees of freedom that will be fixed by physical requirements [37, 40]. $\mathbf{Z}$ reflects the freedom in defining $\mathbf{Y}$ and $\mathbf{W}$ and can be absorbed into either $\mathbf{Y}$ or $\mathbf{W}$ by redefining them as $\mathbf{Y} + \delta\mathbf{Z}$ or $\mathbf{W} + \mathrm{d}\mathbf{Z}$, respectively; $\mathbf{Z}$ represents the $\delta$-exact part of $\mathbf{Y}$ or the d-exact part of $\mathbf{W}$.

The symplectic density $\boldsymbol{\omega}$, is a $(D-1, 2)$-form closed on spacetime, $\mathrm{d}\boldsymbol{\omega} \overset{\circ}{=} 0$, defined as

$$\boldsymbol{\omega} := \delta\boldsymbol{\Theta} = \delta\boldsymbol{\Theta}_{\mathrm{D}} + \mathrm{d}\delta\mathbf{Y}\,. \tag{2.11}$$

Integrating this symplectic density over a codimension-1 surface $\Sigma_r$ yields the symplectic form $\boldsymbol{\Omega}$ [3]

$$\boldsymbol{\Omega} := \int_{\Sigma_r} \boldsymbol{\omega} = \int_{\Sigma_r} \delta\boldsymbol{\Theta}_{\mathrm{D}} + \int_{\partial\Sigma_r} \delta\mathbf{Y}\,. \tag{2.12}$$

The symplectic form $\boldsymbol{\Omega}$ is a closed $(2, 0)$-form, satisfying $\delta\boldsymbol{\Omega} = 0$ and closedness of $\boldsymbol{\omega}$ implies that $\boldsymbol{\Omega}$ does not depend on the integration surface $\Sigma_r$. Additionally, this form consists of contributions from both codimension-1 and codimension-2 parts. Spacetimes with boundaries typically exhibit both bulk and boundary modes, with the bulk modes appearing in the codimension-1 part of the symplectic form and the boundary modes in the codimension-2 part [66, 67]. As (2.12) indicates, $\boldsymbol{\Omega}$ does not depend on the $\mathbf{W}$ and $\mathbf{Z}$ freedoms in the symplectic potential $\boldsymbol{\Theta}$ and the $\mathbf{Y}$-freedom contributes to its codimension-2 part.

## 2.3 Brief review of gauge/gravity correspondence

This subsection provides a concise review of the core elements of the AdS/CFT correspondence, focusing on its foundational framework. We begin by discussing the AdS/CFT duality, often referred to as the GKPW dictionary [2, 3, 68], which establishes a relationship between the partition functions of two distinct $d+1$ dimensional bulk theory and $d$ dimensional boundary theories. The duality asserts that the

---

[2]These two conditions almost completely determine $\boldsymbol{\Theta}_{\mathrm{D}}$, leaving only a finite contribution undetermined. In Section 8, we will explore this remaining freedom, which leads to intrinsic hydrodynamical deformations.

[3]Whenever the integrals are over differential forms, the measure is already incorporated into the definition of the form.

partition functions of the boundary theory (CFT) and the bulk gravitational theory (AdS) are related as follows

$$\mathcal{Z}_{\text{bdry}}\left[\mathcal{J}^i(x)\right] = \mathcal{Z}_{\text{bulk}}\left[\mathcal{J}^i(x)\right]. \tag{2.13}$$

In the boundary theory, the partition function is given by

$$\mathcal{Z}_{\text{bdry}}\left[\mathcal{J}^i(x)\right] = \int D\phi(x) \exp\left[-S_{\text{CFT}}[\phi(x)] - \int_\Sigma \mathrm{d}^d x \sqrt{-\gamma}\, \mathcal{O}_i(x)\, \mathcal{J}^i(x)\right], \tag{2.14}$$

where $\phi(x)$ is a generic field in the boundary CFT on $\Sigma$, and $\mathcal{O}_i(x)$ is a generic local gauge-invariant operator with scaling dimension $\Delta$ and $\mathcal{J}^i(x)$ represents its coupling, which has scaling dimension $d - \Delta$. The bulk theory partition function is

$$\mathcal{Z}_{\text{bulk}}\left[\mathcal{J}^i(x)\right] = \int_{J^i(x,r_\infty)=r_\infty^{d-\Delta}\mathcal{J}^i(x)} DJ^i(x,r)\, \exp(-S_{\text{bulk}}), \tag{2.15}$$

where $J^i(x,r)$ represents the bulk field, and the boundary condition is imposed at $r = r_\infty$ ($\Sigma$) with Dirichlet boundary conditions. The relation between the boundary and bulk quantities is given by:

$$\mathcal{J}^i(x) = r_\infty^{\Delta-d} J^i(x,r_\infty). \tag{2.16}$$

The scaling dimension $\Delta$ of the operator $\mathcal{O}_i(x)$ is related to the mass $m$ of the bulk field via $\Delta(\Delta - d) = m^2\ell^2$. For non-scalar fields, the relation becomes $\Delta(\Delta - d) = m^2\ell^2 + f(s)$, where $f(s)$ depends on the spacetime dimension $d$ and the Lorentz representation of the operator $\mathcal{O}_i$ [68].

**Gauge/Gravity correspondence.** In a special limit of the AdS/CFT duality, where the bulk gravitational theory becomes classical (i.e., when $l_{\text{Pl}}^2 \mathcal{R} \ll 1$, where $l_{\text{Pl}}$ is the Planck length and $\mathcal{R}$ is the typical curvature radius of the background), and the boundary theory is in the planar limit with a large number of degrees of freedom $N \gg 1$, the duality simplifies. This regime is referred to as the *gauge/gravity correspondence*. In this limit, the AdS/CFT duality reduces to:

$$S_{\text{bdry}}[\phi^*] + \int_\Sigma \sqrt{-\gamma}\, \mathcal{J}^i\, \mathcal{O}_i[\phi^*] = S_{\text{bulk}}[\mathcal{J}^i(x)], \tag{2.17}$$

where $\phi^*$ is the solution to the following saddle-point equation:

$$\phi^* = \phi^*[\mathcal{J}^i] \quad \text{such that} \quad \frac{\delta S_{\text{bdry}}[\phi^*]}{\delta\phi} + \int_\Sigma \sqrt{-\gamma}\, \mathcal{J}^i \frac{\delta \mathcal{O}_i[\phi^*]}{\delta\phi} = 0. \tag{2.18}$$

### 2.4 Brief review of Freelance Holography I

As reviewed in the previous subsection, the standard gauge/gravity correspondence is constructed based on the Dirichlet boundary conditions for bulk fields at the AdS boundary, $\Sigma$. In [34], we extended this correspondence to accommodate arbitrary boundary conditions at $\Sigma$. Witten proposed that, in the large-N limit, changes in the boundary conditions of bulk fields correspond to a multi-trace deformation in the boundary theory [9]; see also [11, 14, 15, 23, 24, 29, 69]. We provided a derivation of this proposal in [34]. In this subsection, we review this construction.

It is well known that adding a $W$-term to an action does not affect the bulk equations of motion, but it can modify the bulk boundary conditions. In other words, boundary conditions on the bulk fields can be altered by introducing an additional boundary term, a $W$-term. One can show that this modification on the bulk side corresponds, in the boundary theory, to a deformation of the form

$$S_{\text{bdry}} \quad \rightarrow \quad \bar{S}_{\text{bdry}} = S_{\text{bdry}} + \int_\Sigma \sqrt{-\gamma}\, \mathcal{W}[\mathcal{O}_i], \tag{2.19}$$

where $\mathcal{W}[\mathcal{O}_i]$ denotes a general multi-trace deformation of the boundary theory. This deformation is related to the bulk $W$-term:

$$W_{\mathrm{bulk}}[\mathcal{O}_i, \mathcal{J}^i] = \sqrt{-\gamma}\left(\mathcal{W}[\mathcal{O}_i, \mathcal{J}^i] - \mathcal{J}^i \mathcal{O}_i\right), \tag{2.20}$$

where $W_{\mathrm{bulk}} := W_{\mathrm{bulk}}^r$. For consistency, $\mathcal{J}^i$ must satisfy the equation

$$\mathcal{J}^i = \frac{\delta(\sqrt{-\gamma}\,\mathcal{W})}{\delta(\sqrt{-\gamma}\,\mathcal{O}_i)}, \qquad \delta W_{\mathrm{bulk}} = \left(\frac{\delta(\sqrt{-\gamma}\,\mathcal{W})}{\delta \mathcal{J}^i} - \sqrt{-\gamma}\,\mathcal{O}_i\right)\delta \mathcal{J}^i. \tag{2.21}$$

This is Witten's prescription for bulk boundary conditions when turning on the multi-trace deformation $\mathcal{W}[\mathcal{O}_i]$ in the boundary theory [9].

**CPS freedoms in gauge/gravity correspondence.** One of the key results of in [34] was the complete fixation of both bulk and boundary CPSF freedoms, as required by the gauge/gravity correspondence. This was achieved through the variation of the fundamental duality relation (2.13)

$$\delta \mathcal{Z}_{\mathrm{bdry}}\left[\mathcal{J}^i(x)\right] = \delta \mathcal{Z}_{\mathrm{bulk}}\left[\mathcal{J}^i(x)\right], \tag{2.22}$$

which played a central role in the procedure. Using this relation, we impose the following conditions on the boundary CPS freedoms $(\mathbf{W}_{\mathrm{bdry}}, \mathbf{Y}_{\mathrm{bdry}}, \mathbf{Z}_{\mathrm{bdry}})$:

1. $\mathbf{W}_{\mathrm{bdry}}$ is fixed by/fixes the boundary conditions of the boundary theory, such as the Dirichlet condition.

2. Assuming that $\partial\Sigma$ is boundary-less, we set $\mathbf{Y}_{\mathrm{bdry}}$ and $\mathbf{Z}_{\mathrm{bdry}}$ to zero.

Next, we fix the bulk CPS freedoms $(\mathbf{W}_{\mathrm{bulk}}, \mathbf{Y}_{\mathrm{bulk}}, \mathbf{Z}_{\mathrm{bulk}})$ as follows:

1. $\mathbf{W}_{\mathrm{bulk}}$ is fixed by the boundary conditions of the bulk theory. It is the generator of change of slicings (canonical transformations) on the bulk solution space [70–75] and is associated with multi-trace deformations of the boundary theory.

2. $\mathbf{Z}_{\mathrm{bulk}}$ is determined in terms of $\mathbf{W}_{\mathrm{bdry}}$ by the following simple relation:

$$\mathbf{Z}_{\mathrm{bulk}} = \mathbf{W}_{\mathrm{bdry}}. \tag{2.23}$$

3. Finally, $\mathbf{Y}_{\mathrm{bulk}}$ is determined as

$$\mathbf{Y}_{\mathrm{bulk}} = \mathbf{\Theta}_{\mathrm{bdry}}^{\mathrm{D}} = \mathbf{\Theta}_{\mathrm{bdry}} - \delta\mathbf{W}_{\mathrm{bdry}}. \tag{2.24}$$

In the following sections, we will extend these results to the cases where the boundary theory is defined on a generic timelike codimension-one surface inside the bulk of AdS. Before doing so, however, we need a holographic framework at finite distances, such as the relations in (2.13) and (2.22), to guide us.

## 3 Holography at finite distance with Dirichlet boundary condition

In this section, we systematically extend the gauge/gravity correspondence with Dirichlet boundary conditions to a setting where gravity is confined to a subregion of AdS, bounded by a codimension-one surface $\Sigma_c$, where the boundary theory resides. In our construction, both the bulk theories on $\mathcal{M}$ and $\mathcal{M}_c$ are subject to Dirichlet boundary conditions. Building on this framework, we propose an extension that elevates this correspondence to a full-fledged duality.

### 3.1 Bulk theory bounded to $r \leq r_c$ region inside $\Sigma_c$

Consider a bulk theory in an asymptotically $\text{AdS}_{d+1}$ spacetime, confined to the region $\mathcal{M}_r$, with arbitrary boundary conditions imposed on the boundary $\Sigma_r$. The action for this setup is given by

$$S_{\text{bulk}} = \int_{\mathcal{M}_r} \mathcal{L}_{\text{bulk}}, \qquad \mathcal{L}_{\text{bulk}} = \mathcal{L}_{\text{bulk}}^{\text{D}} + \partial_\mu W_{\text{bulk}}^\mu, \tag{3.1}$$

where, $\mathcal{L}_{\text{bulk}}^{\text{D}}$ represents the bulk Lagrangian compatible with Dirichlet boundary conditions on $\Sigma_r$, while $\mathcal{L}_{\text{bulk}}$ denotes the bulk Lagrangian corresponding to a different set of boundary conditions, specified by the boundary Lagrangian $W_{\text{bulk}}^\mu$. As discussed in the previous section, the boundary term serves to modify the boundary conditions of the bulk theory to arbitrary ones. The on-shell variation of the Lagrangian is then given by

$$\delta \mathcal{L}_{\text{bulk}} \overset{\circ}{=} \partial_\mu \Theta_{\text{bulk}}^\mu, \tag{3.2a}$$

$$\Theta_{\text{bulk}}^\mu := \Theta_{\text{D}}^\mu + \delta W_{\text{bulk}}^\mu + \partial_\nu Y_{\text{bulk}}^{\mu\nu} + \partial_\nu \delta Z_{\text{bulk}}^{\mu\nu}, \tag{3.2b}$$

where $\Theta_{\text{D}}^\mu$ is the symplectic potential compatible with Dirichlet boundary conditions at $\Sigma_r$. The term $\partial_\mu \Theta_{\text{bulk}}^\mu$ can be expressed as follows

$$\begin{aligned}
\partial_\mu \Theta_{\text{bulk}}^\mu &= \partial_r \Theta_{\text{bulk}}^r + \partial_a \Theta_{\text{bulk}}^a = \partial_r \left( \Theta_{\text{bulk}}^r + \partial_a \int^r \mathrm{d}r' \, \Theta_{\text{bulk}}^a \right) \\
&= \partial_r \left( \Theta_{\text{D}}^r + \delta W_{\text{bulk}} + \partial_a Y_{\text{bulk}}^{ra} + \partial_a \delta Z_{\text{bulk}}^{ra} + \partial_a \int^r \mathrm{d}r' \, \Theta_{\text{bulk}}^a \right),
\end{aligned} \tag{3.3}$$

where we have used the definition given in (3.2b). Next, we redefine the $Y$-term to include the last term from the second line in the above equation,

$$\bar{Y}_{\text{bulk}}^{ra} = Y_{\text{bulk}}^{ra} + \int^r \mathrm{d}r' \, \Theta_{\text{bulk}}^a. \tag{3.4}$$

This adjustment of $Y_{\text{bulk}}$ is consistent with the McNees-Zwikel prescription, which ensures finite surface charges [76]. With this modification, where

$$\partial_\mu \Theta_{\text{bulk}}^\mu = \partial_r \left( \Theta_{\text{D}}^r + \delta W_{\text{bulk}} + \partial_a \bar{Y}_{\text{bulk}}^{ra} + \partial_a \delta Z_{\text{bulk}}^{ra} \right), \tag{3.5}$$

integrating (3.2a) over the codimension-one hypersurface $\Sigma_r$ and applying (3.5), we obtain

$$\frac{\mathrm{d}}{\mathrm{d}r} \int_{\Sigma_r} \left( \Theta_{\text{D}}^r + \delta W_{\text{bulk}} + \partial_a \bar{Y}_{\text{bulk}}^{ra} + \partial_a \delta Z_{\text{bulk}}^{ra} \right) = \int_{\Sigma_r} \delta \mathcal{L}_{\text{bulk}}. \tag{3.6}$$

Since $W$-freedom is the generator of canonical transformations, we have

$$\int_{\Sigma_r} \left( \Theta_{\text{D}}^r + \delta W_{\text{bulk}} \right) = \int_{\Sigma_r} \sqrt{-h} \, \tilde{O}_i(x^a, r) \, \delta \tilde{J}^i(x^a, r), \tag{3.7}$$

where $\tilde{J}^i$ and $\tilde{O}_i$ constitute a pair of canonical variables induced by the boundary term $W_{\text{bulk}}$ on $\Sigma_r$. Integrating (3.6) over $r$ from $r_1$ to $r_2$ and using (3.7), yields

$$\begin{aligned}
\delta \int_{r_1}^{r_2} \mathrm{d}r \int_{\Sigma_r} \mathcal{L}_{\text{bulk}} &= \int_{\Sigma_{r_2}} \sqrt{-h} \, \tilde{O}_i \, \delta \tilde{J}^i - \int_{\Sigma_{r_1}} \sqrt{-h} \, \tilde{O}_i \, \delta \tilde{J}^i \\
&\quad + \int_{\partial \Sigma_{r_2}} n_a \left( \bar{Y}_{\text{bulk}}^{ra} + \delta Z_{\text{bulk}}^{ra} \right) - \int_{\partial \Sigma_{r_1}} n_a \left( \bar{Y}_{\text{bulk}}^{ra} + \delta Z_{\text{bulk}}^{ra} \right),
\end{aligned} \tag{3.8}$$

where $n_a = \partial_a(\partial \Sigma_r)$ is a non-normalized vector orthogonal to $\partial \Sigma_r$. The left-hand side represents the on-shell variation of the total action within the spacetime region enclosed by $\Sigma_{r_1}$ and $\Sigma_{r_2}$. This equation will play a central role in the discussions that follow.

## 3.2 Holography at finite cutoff with Dirichlet boundary conditions, bulk theory

Here, we develop holography in the large-N limit at a finite distance $\Sigma_r$, with Dirichlet boundary conditions for the bulk fields. Specifically, we set $W^\mu_{\text{bulk}} = 0$ on $\Sigma_r$ in this subsection, and will introduce it in the subsequent subsections.

Let us begin with the standard gauge/gravity correspondence under the Dirichlet boundary condition, given by

$$S_{\text{bdry}}[\phi^*] + \int_\Sigma \sqrt{-\gamma}\, \mathcal{J}^i \mathcal{O}_i[\phi^*] = S_{\text{bulk}}[\mathcal{J}^i]. \tag{3.9}$$

As mentioned earlier, $\phi^*$ is solution of (2.18) and thus depends on $\mathcal{J}^i$, i.e., $\phi^* = \phi^*[\mathcal{J}^i]$. Consequently, the left-hand side of the equation is also a function of $\mathcal{J}^i$. Next, we vary (3.9) with respect to $\mathcal{J}^i$ while incorporating the saddle point equation (2.18), yielding

$$\int_\Sigma \sqrt{-\gamma}\, \mathcal{O}_i\, \delta \mathcal{J}^i + \int_{\partial\Sigma} n_a\, \Theta^a_{\text{bdry}} = \delta S_{\text{bulk}}[\mathcal{J}^i]. \tag{3.10}$$

The left-hand side of this equation originates from the boundary theory. Using the relations $\sqrt{-\gamma} = r_\infty^{-d}\sqrt{-h}$ and the holographic dictionary $\mathcal{J}^i = r_\infty^{d-\Delta} J^i$, $\mathcal{O}_i = r_\infty^\Delta O_i$, we can rewrite it as

$$\int_\Sigma \sqrt{-h}\, O_i\, \delta J^i + \int_{\partial\Sigma} n_a\, \Theta^a_{\text{bdry}} = \delta S_{\text{bulk}}[J^i; r_\infty]. \tag{3.11}$$

Subtracting (3.8) for $r_2 = \infty$ and $r_1 = r_c$ (where $r = r_c$ is the codimension-one surface the dual boundary theory resides) off (3.11), $O_i\, \delta J^i$ at $r = \infty$ drops out and we obtain an expression which only involves $O_i\, \delta J^i$ at $r_c$:

$$\int_{\Sigma_c} \sqrt{-h}\, O_i\, \delta J^i + \int_{\partial\Sigma_c} n_a\, \left(\bar{Y}^{ra}_{\text{bulk}} + \delta Z^{ra}_{\text{bulk}}\right)$$
$$= \delta\left[S_{\text{bulk}}[J^i; r_\infty] - \int_{r_c}^\infty \mathrm{d}r \int_{\Sigma_r} \mathcal{L}^{\text{D}}_{\text{bulk}}\right] + \int_{\partial\Sigma} n_a\, \left(\bar{Y}^{ra}_{\text{bulk}} + \delta Z^{ra}_{\text{bulk}} - \Theta^a_{\text{bdry}}\right), \tag{3.12}$$

where $\Sigma_c$ denotes the constant $r = r_c$ surface. Recall that $\Theta_{\text{bdry}}$ can be decomposed as $\Theta_{\text{bdry}} = \Theta^{\text{D}}_{\text{bdry}} + \delta W_{\text{bdry}}$. Thus, following the prescriptions of [34] along with (2.23) and (2.24), we observe that the last integral of the second line of the above equation vanishes, resulting in

$$\int_{\Sigma_c} \sqrt{-h}\, O_i\, \delta J^i + \int_{\partial\Sigma_c} n_a\, \left(\bar{Y}^{ra}_{\text{bulk}} + \delta Z^{ra}_{\text{bulk}}\right) = \delta S_{\text{bulk}}[J^i; r_c], \tag{3.13}$$

where $S_{\text{bulk}}[J^i; r_c]$ is the bulk on-shell action up to boundary $\Sigma_c$

$$S_{\text{bulk}}[J^i; r_c] = S_{\text{bulk}}[J^i; r_\infty] - \int_{r_c}^\infty \mathrm{d}r \int_{\Sigma_r} \mathcal{L}^{\text{D}}_{\text{bulk}} = \int_0^{r_c} \mathrm{d}r \int_{\Sigma_r} \mathcal{L}^{\text{D}}_{\text{bulk}}\Big|_{\text{on-shell}} = \int_{\mathcal{M}_c} \mathcal{L}^{\text{D}}_{\text{bulk}}\Big|_{\text{on-shell}}. \tag{3.14}$$

We are now ready to develop holography at $r_c$. To do this, we proceed by constructing it in parallel with the standard gauge/gravity correspondence. As in the standard gauge/gravity framework, we assume the existence of a quantum field theory defined on a cutoff surface $\Sigma_c$. We then construct its $W$-freedom and symplectic potential in a manner analogous to the expressions in (2.23) and (2.24), yielding the following relations

$$\mathbf{Z}_{\text{bulk}} = \mathbf{W}_{\text{bdry}} \quad \text{and} \quad \bar{\mathbf{Y}}_{\text{bulk}} = \boldsymbol{\Theta}^{\text{D}}_{\text{bdry}}, \quad \text{on} \quad \partial\Sigma_c. \tag{3.15}$$

From here, we obtain

$$\Theta^a_{\text{bdry}} = \bar{Y}^{ra}_{\text{bulk}} + \delta Z^{ra}_{\text{bulk}}, \quad \text{on} \quad \partial\Sigma_c. \tag{3.16}$$

Substituting this equation into (3.13), we obtain

$$\int_{\Sigma_c} \sqrt{-h}\, O_i\, \delta J^i + \int_{\partial\Sigma_c} n_a \Theta^a_{\mathrm{bdry}} = \delta S_{\mathrm{bulk}}[J^i; r_c]\,. \tag{3.17}$$

We then proceed in a way similar to the procedure used at infinity. We rescale the quantities defined on $\Sigma_c$ in (3.17) as follows

$$\mathcal{J}^i = r_c^{d-\Delta} J^i(x^a, r_c)\,, \qquad \mathcal{O}_i = r_c^{\Delta} O_i(x^a, r_c)\,, \qquad \sqrt{-\gamma} = r_c^{-d}\sqrt{-h(x^a, r_c)}\,, \tag{3.18}$$

and express the left-hand side of (3.17) in terms of these rescaled quantities intrinsic to $\Sigma_c$

$$\delta S_{\mathrm{bulk}}[J^i; r_c] = \int_{\Sigma_c} \sqrt{-\gamma}\, \mathcal{O}_i\, \delta\mathcal{J}^i + \int_{\partial\Sigma_c} n_a \Theta^a_{\mathrm{bdry}}\,. \tag{3.19}$$

Comparing the above with its counterpart at infinity, given by (3.11), we observe that the structure remains identical, with two key differences: 1) $S_{\mathrm{bulk}}[J^i; r_c]$ represents the on-shell bulk action up to the cutoff $r_c$, rather than extending to infinity. 2) The right-hand side of (3.19) involves integrals over the cutoff surface $\Sigma_c$ and its corner $\partial\Sigma_c$, instead of the asymptotic boundary of AdS, $\Sigma$, and its corner $\partial\Sigma$.

The argument above suggests the following expression for the gauge/gravity correspondence at a finite cutoff:

$$\boxed{S^c_{\mathrm{bdry}}[\phi^*[\mathcal{J}^i]] + \int_{\Sigma_c} \sqrt{-\gamma}\, \mathcal{J}^i\, \mathcal{O}_i[\phi^*[\mathcal{J}^i]] = S_{\mathrm{bulk}}[J^i; r_c]\,,} \tag{3.20}$$

where $S^c_{\mathrm{bdry}}$ is the action of the boundary theory on $\Sigma_c$, and $\phi^*[\mathcal{J}^i]$ is the solution to the saddle point equation

$$\phi^* = \phi^*[\mathcal{J}^i] \quad \text{such that} \quad \frac{\delta S^c_{\mathrm{bdry}}[\phi^*]}{\delta\phi} + \int_{\Sigma_c} \sqrt{-\gamma}\, \mathcal{J}^i \frac{\delta \mathcal{O}_i[\phi^*]}{\delta\phi} = 0\,. \tag{3.21}$$

In essence, equation (3.20) generalizes equation (3.9) to accommodate an arbitrary timelike boundary at $r = r_c$. The next step is to determine $S^c_{\mathrm{bdry}}$, which will be addressed in the following subsection.

## 3.3 Boundary theory at $\Sigma_c$ as a deformation of the boundary theory at $\Sigma$

To complete the construction of holography at a finite distance, we need to explicitly determine the boundary theory at $\Sigma_c$. In this subsection, we demonstrate that the boundary theory at $\Sigma_c$ is a quantum field theory (QFT) that arises as a deformation of the boundary theory at the AdS boundary, $\Sigma$. We then derive the deformation flow equation, which relates the two boundary theories at $\Sigma$ and $\Sigma_c$. To achieve this, we explore their relationships using two different approaches. In the first approach, we start with the correspondence at $\Sigma_c$ given in (3.20), which is

$$S^c_{\mathrm{bdry}}[\phi^*[\mathcal{J}^i]] + \int_{\Sigma_c} \sqrt{-\gamma}\, \mathcal{J}^i\, \mathcal{O}_i[\phi^*[\mathcal{J}^i]] = S_{\mathrm{bulk}}[J^i; r_c]\,, \tag{3.22}$$

where $S^c_{\mathrm{bdry}}[\phi^*[\mathcal{J}^i]]$ is the action of the boundary QFT defined on $\Sigma_r$. To proceed, we directly compare equations (3.9) and (3.20) in order to examine the theories at finite $r_c$ and $r_\infty$, as follows

$$S_{\mathrm{bulk}}[J^i; r_\infty] - S_{\mathrm{bulk}}[J^i; r_c] = S_{\mathrm{bdry}}[\mathcal{J}^i] - S^c_{\mathrm{bdry}}[\mathcal{J}^i] + \int_{\Sigma} \sqrt{-\gamma}\, \mathcal{J}^i\, \mathcal{O}_i[\mathcal{J}^i] - \int_{\Sigma_c} \sqrt{-\gamma}\, \mathcal{J}^i\, \mathcal{O}_i[\mathcal{J}^i]\,. \tag{3.23}$$

Employing the following definitions,

$$\hat{S}_{\text{bdry}}[\mathcal{J}^i] := S_{\text{bdry}}[\mathcal{J}^i] + \int_{\Sigma} \sqrt{-\gamma}\, \mathcal{J}^i\, \mathcal{O}_i[\mathcal{J}^i]\,,$$

$$\hat{S}^{\text{c}}_{\text{bdry}}[\mathcal{J}^i] := S^{\text{c}}_{\text{bdry}}[\mathcal{J}^i] + \int_{\Sigma_c} \sqrt{-\gamma}\, \mathcal{J}^i\, \mathcal{O}_i[\mathcal{J}^i]\,, \tag{3.24}$$

and (3.14), the expression (3.23) simplifies to

$$\hat{S}^{\text{c}}_{\text{bdry}} = \hat{S}_{\text{bdry}} - \int_{r_c}^{\infty} dr\, \mathcal{S}_{\text{deform}}(r)\,, \qquad \mathcal{S}_{\text{deform}}(r) := \int_{\Sigma_r} \mathcal{L}^{\text{D}}_{\text{bulk}}\Big|_{\text{on-shell}}\,. \tag{3.25}$$

Equivalently, we obtain the deformation flow equation:

$$\boxed{\frac{d}{dr}\hat{S}_{\text{bdry}}(r) = \mathcal{S}_{\text{deform}}(r)\,,} \tag{3.26}$$

where $\hat{S}_{\text{bdry}}(r \to \infty) = \hat{S}_{\text{bdry}}$ and $\hat{S}_{\text{bdry}}(r \to r_c) = \hat{S}^{\text{c}}_{\text{bdry}}$. This represents the deformation flow equation, where $r_c$ acts as the deformation parameter. In the following sections, we will derive the explicit form of the deformation action for various cases.

**Geometric derivation.**   Now we present another derivation for the deformation flow equation (3.26), using spacetime diffeomorphisms. The variation of the gravitational action under the generic diffeomorphisms $\xi = \xi^\mu \partial_\mu$ is

$$\delta_\xi S_{\text{bulk}}(r) = \int_{\Sigma_r} \sqrt{-g}\, s_\mu\, \xi^\mu\, \mathcal{L}^{\text{D}}_{\text{bulk}}\,, \tag{3.27}$$

where $S_{\text{bulk}}(r)$ is the bulk action on $\mathcal{M}_r$. If we focus on the generator that takes us in the radial direction, $\xi = \partial_r$, then we find

$$\frac{d}{dr}S_{\text{bulk}}(r) = \int_{\Sigma_r} \mathcal{L}^{\text{D}}_{\text{bulk}}\,. \tag{3.28}$$

By evaluating this equation on-shell and using the saddle point approximation, $S_{\text{bulk}}(r) = \hat{S}_{\text{bdry}}(r)$, we obtain

$$\frac{d}{dr}\hat{S}_{\text{bdry}}(r) = \int_{\Sigma_r} \mathcal{L}^{\text{D}}_{\text{bulk}}\Big|_{\text{on-shell}} = \mathcal{S}_{\text{deform}}\,, \tag{3.29}$$

recovering (3.26).

To complete the discussion, we need to express $\mathcal{S}_{\text{deform}}$ in terms of the boundary theory variables. In this regard, we note that the on-shell variation of the gravitational action takes the following form

$$\delta_\xi S_{\text{bulk}}(r) \doteq \int_{\mathcal{M}_r} \partial_\mu \Theta^\mu_{\text{bulk}}[\delta_\xi \mathcal{J}^i] = \int_{\Sigma_r} \sqrt{-h}\, \mathcal{O}_i\, \delta_\xi \mathcal{J}^i + \int_{\partial\Sigma_r} n_a \left( \bar{Y}^{ra}_{\text{bulk}}[\delta_\xi \mathcal{J}^i] + \delta_\xi Z^{ra}_{\text{bulk}} \right)\,, \tag{3.30}$$

where we have used (3.5). Taking $\xi = \partial_r$ and using (3.16), we get

$$\frac{d}{dr}S_{\text{bulk}}(r) = \int_{\Sigma_r} \sqrt{-h}\, \mathcal{O}_i\, \partial_r \mathcal{J}^i + \int_{\partial\Sigma_r} n_a\, \Theta^a_{\text{bdry}}[\partial_r \phi^*]\,. \tag{3.31}$$

From this, we conclude that

$$\boxed{\mathcal{S}_{\text{deform}}(r_c) = \int_{\Sigma_c} \sqrt{-\gamma}\, \mathcal{O}_i\, r_c^{d-\Delta}\, \partial_{r_c}\left( r_c^{-d+\Delta}\, \mathcal{J}^i \right) + \int_{\partial\Sigma_c} n_a\, \Theta^a_{\text{bdry}}[\partial_{r_c}\phi^*]\,,} \tag{3.32}$$

where we have used the rescaled quantities (3.18) to rewrite the bulk quantities in terms of boundary variables. Also note that $\partial_r J^i$ is related to the canonical momenta associated with $J^i$ over $\Sigma_r$, namely, $O_i$. However, the explicit form of these deformations depends on the specific bulk theory, which we will explore in more detail in the following sections.

We conclude this section by discussing tangential diffeomorphisms on $\Sigma_r$. For $s_\mu \xi^\mu = 0$, (3.27) yields

$$\delta_\xi \hat{S}_{\text{bdry}}(r) = 0 \,. \tag{3.33}$$

This implies that a subset of bulk diffeomorphisms, specifically those tangential to $\Sigma_r$, correspond to symmetries of the boundary QFT action. From the bulk perspective, these diffeomorphisms generate nonzero gravitational surface charges, rendering them physically relevant. A notable example is the Brown-Henneaux diffeomorphisms [77], which belong to this class of tangential transformations. Since the charges associated with tangential diffeomorphisms are captured by codimension-two integrals in the bulk (often referred to as surface charges), they manifest as codimension-one integrals when viewed from the boundary perspective. As a result, they correspond to global symmetries of the boundary theory. From this, we deduce that for $s_\mu \xi^\mu = 0$, the following must hold

$$\int_{\Sigma_c} \sqrt{-\gamma}\, O_i \, \mathcal{L}_\xi \mathcal{J}^i + \int_{\partial\Sigma_c} n_a \left( \bar{Y}^{ra}_{\text{bulk}}[\delta_\xi J^i] + \delta_\xi Z^{ra}_{\text{bulk}} \right) = 0 \,. \tag{3.34}$$

This equation must hold for any tangential diffeomorphism, and it allows us to derive the Ward identities on the QFT side. In the following sections, we will examine several examples that illustrate this point.

**AdS/QFT duality.** At this stage, we have constructed holography at a finite distance in the large-N limit, directly from the gauge/gravity correspondence at infinity. We now propose a conjectural holographic duality on $\Sigma_c$

$$\boxed{\left\langle \exp\left( \int_{\Sigma_c} \sqrt{-\gamma}\, O_i(x)\, \mathcal{J}^i(x) \right) \right\rangle_{\text{QFT}} = \mathcal{Z}_{\text{bulk}}\left[ J^i(x, r_c) = r_c^{\Delta-d}\, \mathcal{J}^i \right] \,.} \tag{3.35}$$

This proposal is the same as the one introduced in [56]. In this framework, (3.20) naturally emerges as the saddle-point approximation of the duality (3.35). Finally, we note that, similar to the standard AdS/CFT prescription at the asymptotic boundary, we rescaled the bulk fields with appropriate powers of $r_c$ as shown in (3.18). Although this rescaling is essential at infinity, it is not strictly necessary at a finite cutoff, as discussed in [56]. However, we adopted this minimal rescaling at a finite distance to ensure that, at the limit $r_c \to \infty$, we recover the standard results.

## 4 Holography at finite distance with arbitrary boundary conditions

In the previous section, we developed holography at a finite distance with Dirichlet boundary conditions. In this section, we extend the framework to arbitrary boundary conditions at an arbitrary cutoff $r_c$. The construction follows the same approach as in [34] (cf. subsection 2.4). We begin with

$$S^c_{\text{bdry}}[\phi^*[\mathcal{J}^i]] + \int_{\Sigma_c} \sqrt{-\gamma}\, \mathcal{J}^i \, O_i[\phi^*[\mathcal{J}^i]] = S_{\text{bulk}}[J^i; r_c] \,, \tag{4.1}$$

and introduce a multi-trace deformation by adding $\int_{\Sigma_c} \sqrt{-\gamma}\, \mathcal{W}[O_i[\mathcal{J}^i], \mathcal{J}^i]$ to both sides. This allows us to rewrite the equation as

$$\bar{S}^c_{\text{bdry}}[\phi^*[\mathcal{J}^i]] = \bar{S}_{\text{bulk}}[\mathcal{J}^i; r_c] \,, \tag{4.2}$$

where following definitions are used

$$\bar{S}_{\text{bdry}}^{\text{c}}[\phi] = S_{\text{bdry}}^{\text{c}}[\phi] + \int_{\Sigma_c} \sqrt{-\gamma}\, \mathcal{W}[\mathcal{O}_i, \mathcal{J}^i]\,,$$

$$\bar{S}_{\text{bulk}}[\mathcal{J}^i; r_c] = S_{\text{bulk}}[\mathcal{J}^i; r_c] + \int_{\Sigma_c} W_{\text{bulk}}[\mathcal{O}_i, \mathcal{J}^i]\,,$$

(4.3)

in which we introduced

$$W_{\text{bulk}}[\mathcal{O}_i, \mathcal{J}^i] = \sqrt{-\gamma}\left(\mathcal{W}[\mathcal{O}_i, \mathcal{J}^i] - \mathcal{J}^i\,\mathcal{O}_i\right)\,.$$

(4.4)

This demonstrates that multi-trace deformations of the boundary QFT correspond to $W_{\text{bulk}}$ freedom in the bulk theory. The consistency condition further implies

$$\mathcal{J}^i = \frac{\delta(\sqrt{-\gamma}\,\mathcal{W})}{\delta(\sqrt{-\gamma}\,\mathcal{O}_i)}\,.$$

(4.5)

In summary, in subsection 3.2, we constructed a duality at the cutoff surface $\Sigma_c$ with Dirichlet boundary conditions for the bulk fields. In this subsection, leveraging the $W_{\text{bulk}}$ freedom, we extend the holographic framework to accommodate arbitrary boundary conditions on $\Sigma_c$. We demonstrated that achieving this requires deforming the boundary QFT through multi-trace deformations induced by the $W_{\text{bulk}}$ freedom.

## 5   Interpolating boundary conditions between two boundaries

In section 3, we formulated holography at a finite distance with Dirichlet boundary conditions, and in section 4, we extend this framework to arbitrary boundary conditions. Here, we develop the formulation that allows us to interpolate between different boundary conditions at different radii, thereby completing the program of *Freelance Holography II*. First, we analyze the interpolation between two arbitrary radii, with the Dirichlet boundary condition applied to one of them. Subsequently, we extend this approach to interpolation between two arbitrary boundary conditions.

### 5.1   Evolving from a given boundary condition

To explore the interpolation of boundary conditions between two infinitesimally close boundaries, $\Sigma_r$ and $\Sigma_{r+\text{d}r}$, as illustrated in Fig. 2. We start with (3.6) and (3.7),

$$\frac{\text{d}}{\text{d}r}\left[\int_{\Sigma_r}\left(\sqrt{-h}\,\tilde{O}_i\,\delta\tilde{J}^i\right) + \int_{\partial\Sigma_r} n_a\left(\bar{Y}_{\text{bulk}}^{ra} + \delta Z_{\text{bulk}}^{ra}\right)\right] = \int_{\Sigma_r}\delta\mathcal{L}_{\text{bulk}}\,.$$

(5.1)

The second term on the left-hand side is a corner term and is not essential to the discussion of boundary conditions and boundary deformation. Thus, we omit its explicit form. From the equation above, we obtain

$$\int_{\Sigma_{r+\text{d}r}}\sqrt{-h}\,\tilde{O}_i\,\delta\tilde{J}^i = \int_{\Sigma_r}\sqrt{-h}\,\tilde{O}_i\,\delta\tilde{J}^i + \text{d}r\int_{\Sigma_r}\delta\mathcal{L}_{\text{bulk}} + \text{corner terms}\,.$$

(5.2)

To discuss the physical significance of the above equation, we first recall that setting $\delta\tilde{J}^i = 0$ on $\Sigma_r$ or $\Sigma_{r+\delta r}$ corresponds to imposing a particular type of boundary condition, depending on the employed $W$-freedom. These boundary conditions can include Dirichlet, Neumann, conformal, or other conditions. We can absorb the total variation term on the right-hand side of (5.2) into the first term and rewrite the equation as

$$\int_{\Sigma_{r+\text{d}r}}\sqrt{-h}\,\tilde{O}_i\,\delta\tilde{J}^i = \int_{\Sigma_r}\sqrt{-h}\,\tilde{\tilde{O}}_i\,\delta\tilde{\tilde{J}}^i + \text{corner terms}\,.$$

(5.3)

This equation explicitly shows that the same type of boundary conditions cannot be imposed simultaneously at both infinitesimally close boundaries $\Sigma_{r+\mathrm{d}r}$ and $\Sigma_r$. In particular, if a specific boundary condition is imposed at $\Sigma_{r+\mathrm{d}r}$, such as $\delta \tilde{J}^i = 0$, then the boundary conditions at $\Sigma_r$ are not independent but are instead automatically determined, $\delta \tilde{\tilde{J}}^i = 0$,

$$\sqrt{-h}\, \tilde{O}_i \, \frac{\delta \tilde{\tilde{J}}^i}{\delta O_i} = \mathrm{d}r \frac{\delta \mathcal{L}_{\text{bulk}}}{\delta O_i} \,, \qquad \sqrt{-h}\, \tilde{O}_i \, \frac{\delta \tilde{\tilde{J}}^i}{\delta O_i} = \sqrt{-h}\, \tilde{O}_i + \mathrm{d}r \frac{\delta \mathcal{L}_{\text{bulk}}}{\delta J^i} \,. \tag{5.4}$$

As a specific example, consider $\Sigma_r \to \Sigma$ and $\Sigma_{r+\mathrm{d}r} \to \Sigma_c$ with $\mathrm{d}r < 0$. The above demonstrates how the boundary conditions should evolve as we move infinitesimally from the AdS boundary into the bulk. One can rewrite (5.2) in the following form

$$\boxed{\int_{\Sigma_{r+\mathrm{d}r}} \sqrt{-h}\, \tilde{O}_i \, \delta \tilde{J}^i = \int_{\Sigma_r} \sqrt{-h}\, \tilde{O}_i \, \delta \tilde{J}^i + \mathrm{d}r\, \delta \mathcal{S}_{\text{deform}}^{\text{W}}(r) = \int_{\Sigma_r} \sqrt{-h}\, \tilde{O}_i \, \delta \tilde{\tilde{J}}^i \,,} \tag{5.5}$$

where we have omitted the corner terms and

$$\mathcal{S}_{\text{deform}}^{\text{W}}(r) = \int_{\Sigma_r} \mathcal{L}_{\text{bulk}} = \int_{\Sigma_r} \left( \mathcal{L}_{\text{bulk}}^{\text{D}} + \partial_\mu W_{\text{bulk}}^\mu \right) \,. \tag{5.6}$$

This represents the extension of (3.25) for the transition between two boundaries at different radii with a given boundary condition. In the Dirichlet case, $W_{\text{bulk}}^\mu = 0$, (5.6) reduces to (3.25).

This equation, in accord with T$\bar{\text{T}}$-deformation literature [42, 65], admits two possible interpretations:

I. **Radial evolution:** Consider the first equality in (5.5). The term $\int_{\Sigma_r} \sqrt{-h}\, \tilde{O}_i \, \delta \tilde{J}^i$ on the right-hand side is consistent with the boundary condition $\delta \tilde{J}^i \big|_{\Sigma_r} = 0$. Adding the term $\mathrm{d}r\, \delta \mathcal{S}_{\text{deform}}^{\text{W}}$ results in the symplectic potential on $\Sigma_{r+\mathrm{d}r}$ (left-hand side), which remains compatible with the boundary condition $\delta \tilde{J}^i \big|_{\Sigma_{r+\mathrm{d}r}} = 0$. This condition is of the same type as that imposed on $\Sigma_r$ before the addition of $\mathrm{d}r\, \delta \mathcal{S}_{\text{deform}}^{\text{W}}$. Therefore, if we begin with a bulk symplectic potential of type X on $\Sigma$ and modify it by adding $\mathcal{S}_{\text{deform}}^{\text{W}}$, the resulting symplectic potential on $\Sigma_{r+\mathrm{d}r}$ will still satisfy the same type of boundary condition X. Conversely, the first equality in (5.5) shows that as we move from $\Sigma_{r+\mathrm{d}r}$ to $\Sigma_r$ one can keep the same boundary conditions IFF $\delta \mathcal{S}_{\text{deform}}^{\text{W}}$ vanishes at $r$.

II. **Boundary condition evolution:** We now consider the second equality in (5.5), which is an equality of two integrals over $\Sigma_r$. The term $\int_{\Sigma_r} \sqrt{-h}\, \tilde{O}_i \, \delta \tilde{J}^i$ represents the on-shell symplectic potential on $\Sigma_r$, which is consistent with the boundary condition $\delta \tilde{J}^i \big|_{\Sigma_r} = 0$. By adding the term $\delta \mathcal{S}_{\text{deform}}^{\text{W}}$, we introduce a new symplectic potential on $\Sigma_r$, incorporating a different type of boundary condition. This implies that if we start with a symplectic potential on $\Sigma_r$ satisfying boundary condition type X and then modify it by adding $\delta \mathcal{S}_{\text{deform}}^{\text{W}}$, the resulting symplectic potential on $\Sigma_r$ will instead satisfy a different boundary condition, say type Y, which corresponds to $\delta \tilde{\tilde{J}}^i \big|_{\Sigma_r} = 0$.

We conclude this subsection with an important remark: In item I, we noted that $\delta \mathcal{S}_{\text{deform}}^{\text{W}}$ interpolates between symplectic potentials that satisfy the same boundary condition, X, on $\Sigma_r$ and $\Sigma_{r+\delta r}$. The distinction between different choices of boundary condition X becomes clear from the explicit form of $\mathcal{S}_{\text{deform}}^{\text{W}}$ in (5.6), where different boundary conditions correspond to different choices of the $W$-term.

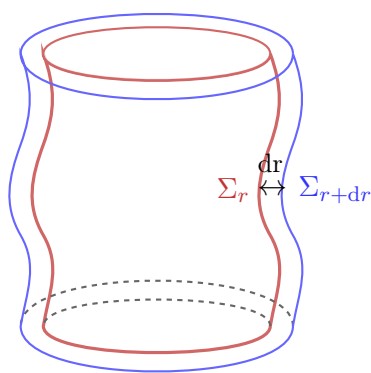

**Figure 2**: Two infinitesimally close boundaries $\Sigma_r$ and $\Sigma_{r+\mathrm{d}r}$ in asymptotically AdS spacetimes.

## 5.2 Interpolating between two arbitrary boundary conditions at different radii

We now complete the freelance holography program by constructing the interpolation between symplectic potentials at different radii for two *arbitrary* boundary conditions. In other words, we examine the transition from a boundary condition of type X to one of type Y at different radial slices. Equation (5.5) describes the transition between boundary conditions of the same type. To generalize it to arbitrary boundary conditions, we start from (5.5) and (5.6)

$$\int_{\Sigma_{r+\mathrm{d}r}} \sqrt{-h}\, \tilde{O}_i\, \delta \tilde{J}^i = \int_{\Sigma_r} \sqrt{-h}\, \tilde{O}_i\, \delta \tilde{J}^i + \mathrm{d}r\, \delta \int_{\Sigma_r} \mathcal{L}_{\text{bulk}}\,, \tag{5.7}$$

and rewrite it as follows

$$\int_{\Sigma_{r+\mathrm{d}r}} \sqrt{-h}\, \tilde{O}_i\, \delta \tilde{J}^i = \int_{\Sigma_r} \sqrt{-h}\, \tilde{O}_i\, \delta \tilde{J}^i + \mathrm{d}r\, \delta \int_{\Sigma_r} \mathcal{L}_{\text{bulk}} + \delta \int_{\Sigma_r} W_{Y \to X} - \delta \int_{\Sigma_r} W_{Y \to X}\,. \tag{5.8}$$

where we defined the boundary condition $\delta \tilde{J}^i = 0$ as boundary condition type Y. Since $W$ is the generator of change of slicing, we obtain

$$\int_{\Sigma_r} \sqrt{-h}\, \tilde{O}_i\, \delta \tilde{J}^i + \delta \int_{\Sigma_r} W_{Y \to X} = \int_{\Sigma_r} \sqrt{-h}\, \bar{O}_i \delta \bar{J}^i\,, \tag{5.9}$$

in which we identified the condition $\delta \bar{J}^i = 0$ as boundary condition type X. This equation shows that $W_{Y \to X}$ acts as a generator that transforms the boundary condition from type Y to type X on $\Sigma_r$. Substituting (5.9) into (5.8), we arrive at

$$\int_{\Sigma_{r+\mathrm{d}r}} \sqrt{-h}\, \tilde{O}_i\, \delta \tilde{J}^i = \int_{\Sigma_r} \sqrt{-h}\, \bar{O}_i \delta \bar{J}^i + \mathrm{d}r\, \delta \int_{\Sigma_r} \mathcal{L}_{\text{bulk}} - \delta \int_{\Sigma_r} W_{Y \to X}\,. \tag{5.10}$$

To simplify the interpretation of this equation, we define the following generator

$$\mathcal{S}_{X \to Y}(\Sigma_r) = \mathrm{d}r \int_{\Sigma_r} \mathcal{L}_{\text{bulk}} + \int_{\Sigma_r} W_{X \to Y}\,, \tag{5.11}$$

where we have used $W_{X \to Y} = -W_{Y \to X}$. With this definition, (5.10) can be rewritten as

$$\boxed{\int_{\Sigma_{r+\mathrm{d}r}} \sqrt{-h}\, \tilde{O}_i\, \delta \tilde{J}^i = \int_{\Sigma_r} \sqrt{-h}\, \bar{O}_i \delta \bar{J}^i + \delta \mathcal{S}_{X \to Y}(\Sigma_r) = \int_{\Sigma_r} \sqrt{-h}\, \tilde{\tilde{O}}_i\, \delta \tilde{\tilde{J}}^i\,,} \tag{5.12}$$

where in the second equality, we have used (5.5). As in Section 5.1, this equation admits two interpretations:

I. **Radial evolution:** The first equality in (5.12) shows that $\mathcal{S}_{X \to Y}(\Sigma_r)$ governs the transition from boundary condition type X on $\Sigma_r$ to type Y on $\Sigma_{r+\mathrm{d}r}$. Hence, $\mathcal{S}_{X \to Y}(\Sigma_r)$ serves as the generator of the transition between two different radii, each with arbitrary boundary conditions.

II. **Boundary condition evolution:** The second equality suggests an alternative perspective, where $\mathcal{S}_{X \to Y}(\Sigma_r)$ generates a transformation of boundary conditions over $\Sigma_r$, changing them from type X to another type defined by $\delta \tilde{\tilde{J}}^i|_{\Sigma_r} = 0$.

## 6 Radial evolution from Dirichlet boundary condition, examples

In this section, we demonstrate how the general formulation discussed in Section 5.1 works in practice by working out some different examples and explicitly identifying the form of the deformation that extends holography from the AdS boundary with Dirichlet boundary conditions to a finite-cutoff boundary while preserving Dirichlet boundary conditions. In the next section, we will explore transitions between different boundary conditions.

### 6.1 GR plus matter in various dimensions

In General Relativity with a negative cosmological constant, the bulk action associated with the Dirichlet boundary condition on $\Sigma_r$ is given by

$$S = \int_{\mathcal{M}_r} \mathcal{L}^{\mathrm{D}}_{\mathrm{bulk}} = \frac{1}{2\kappa} \int_{\mathcal{M}_r} \sqrt{-g} \, (R - 2\Lambda + 2\kappa \, \mathcal{L}_{\mathrm{M}}) + \frac{1}{\kappa} \int_{\Sigma_r} \left( \sqrt{-h} \, K + \mathcal{L}_{\mathrm{ct}} \right) , \tag{6.1}$$

where $\kappa := 8\pi G$, and $\mathcal{L}_{\mathrm{M}}$ denotes the matter Lagrangian. The final term includes the Gibbons-Hawking-York (GHY) boundary term [78, 79] along with counterterms [80]. Notably, when the boundary is at a finite distance, the on-shell action and other physical quantities remain finite, eliminating the need for counterterms in a finite-cutoff formulation of holography. However, we include the counterterm Lagrangian $\mathcal{L}_{\mathrm{ct}}$ to ensure a finite on-shell action in the asymptotic $r \to \infty$ limit. Its explicit form depends on the spacetime dimension [80],

$$d = 2 : \qquad \mathcal{L}_{\mathrm{ct}} = -\frac{1}{\ell}\sqrt{-h} \, ,$$

$$d = 3 : \qquad \mathcal{L}_{\mathrm{ct}} = -\frac{2}{\ell}\sqrt{-h} \left( 1 + \frac{\ell^2}{4}\hat{R} \right) , \tag{6.2}$$

$$d = 4 : \qquad \mathcal{L}_{\mathrm{ct}} = -\frac{3}{\ell}\sqrt{-h} \left( 1 + \frac{\ell^2}{12}\hat{R} \right) ,$$

where $\hat{R}$ is the Ricci scalar of $h_{ab}$ and $\Lambda = -\frac{d(d-1)}{2\ell^2}$. The resulting equations of motion are

$$R_{\mu\nu} - \frac{1}{2}R \, g_{\mu\nu} + \Lambda g_{\mu\nu} = \kappa \, T_{\mu\nu} \, , \tag{6.3}$$

where $T_{\mu\nu}$ denotes the matter energy-momentum tensor. The renormalized Brown-York energy-momentum tensor (rBY-EMT), also known as the holographic energy-momentum tensor [80], for general relativity is given as

$$\mathcal{T}^{ab} = \mathring{\mathcal{T}}^{ab} + \mathcal{T}^{ab}_{\mathrm{ct}} \, , \tag{6.4}$$

where the standard Brown-York energy-momentum tensor (BY-EMT) $\mathring{\mathcal{T}}^{ab}$ [81] and the counterterm $\mathcal{T}^{ab}_{\mathrm{ct}}$ are respectively defined through

$$\mathring{\mathcal{T}}^{ab} := \frac{1}{\kappa}(K^{ab} - K h^{ab}) \, , \qquad \mathcal{T}^{ab}_{\mathrm{ct}} := -\frac{1}{\kappa}\frac{2}{\sqrt{-h}}\frac{\delta \mathcal{L}_{\mathrm{ct}}}{\delta h_{ab}} \, . \tag{6.5}$$

The explicit form of EMT counterterm for various spacetime dimensions are as follows

$$d = 2 : \qquad \kappa \, \mathcal{T}_{\text{ct}}^{ab} = \frac{1}{\ell} h^{ab} \, ,$$

$$d = 3 : \qquad \kappa \, \mathcal{T}_{\text{ct}}^{ab} = \frac{2}{\ell} h^{ab} - \ell \hat{G}^{ab} \, , \tag{6.6}$$

$$d = 4 : \qquad \kappa \, \mathcal{T}_{\text{ct}}^{ab} = \frac{3}{\ell} h^{ab} - \frac{\ell}{2} \hat{G}^{ab} \, ,$$

where $\hat{G}_{ab}$ is the Einstein tensor of $h_{ab}$. It is worth mentioning that the counter-term EMT, the choice of the regularization scheme, is fixed upon two requirements: (1) invariance under diffeomorphism at the boundary which implies divergence-free condition $\hat{\nabla}_a \mathcal{T}_{\text{ct}}^{ab} = 0$; (2) respecting the Dirichlet boundary condition. Among other things, these requirements imply that $\mathcal{T}_{\text{ct}}^{ab}$ up to order $\ell$ (in an expansion in powers of $\ell$) is only a function of $h_{ab}$ and $\hat{G}_{ab}$ and not on $K_{ab}$. Furthermore, the expressions in (6.6) are exact in lower dimensions ($d < 5$), whereas in higher dimensions ($d > 4$), additional terms appear [82].

Using our foliation (2.1), $(1 + d)$-decomposed Einstein equations in terms of the BY-EMT are

$$\hat{R} - 2\Lambda + \kappa^2 \mathring{\mathcal{T}}^{ab} \mathring{\mathcal{T}}_{ab} - \frac{\kappa^2 \mathring{\mathcal{T}}^2}{d-1} = -2\kappa \, T_{ss} \, , \tag{6.7a}$$

$$\hat{\nabla}_b \mathring{\mathcal{T}}_a^b = T_{sa} \, , \tag{6.7b}$$

$$\frac{\kappa}{N} \mathcal{D}_r \mathring{\mathcal{T}}_{ab} - 2\kappa^2 \, \mathring{\mathcal{T}}_a^c \mathring{\mathcal{T}}_{bc} + \kappa^2 \frac{\mathring{\mathcal{T}} \mathring{\mathcal{T}}_{ab}}{d-1} - \kappa^2 \left( \mathring{\mathcal{T}}^{ab} \mathring{\mathcal{T}}_{ab} - \frac{\mathring{\mathcal{T}}^2}{d-1} \right) h_{ab}$$

$$+ \frac{\hat{\nabla}_a \hat{\nabla}_b N}{N} - \frac{\hat{\nabla}_c \hat{\nabla}^c N}{N} h_{ab} - \hat{R}_{ab} = -\kappa \, T_{ab} \, , \tag{6.7c}$$

where $\hat{\nabla}$ and $\hat{R}_{ab}$ are the covariant derivative and Ricci tensor associated with $h_{ab}$ on $\Sigma_r$, respectively. We also define the projected components of the matter energy-momentum tensor as $T_{ss} := s^\mu s^\nu T_{\mu\nu}$ and $T_{sa} := s^\mu T_{\mu a}$. While we expressed the Einstein equation in terms of the BY-EMT in (6.7), it can be readily rewritten in terms of the rBY-EMT (6.4).

We now proceed to compute the deformation action (3.32), given by

$$\mathcal{S}_{\text{deform}} = -\frac{1}{2} \int_{\Sigma_r} \sqrt{-h} \, \mathcal{T}^{ab} \, \partial_r h_{ab} + \sum_{i \in \text{matter}} \int_{\Sigma_r} \sqrt{-h} \, O_i \, \partial_r J^i + \text{corner terms} \, . \tag{6.8}$$

Since corner terms at $\partial \Sigma_r$ do not affect the equations of motion of the boundary-deformed theory, hereafter we will omit the contribution of these terms in the calculation of radial deformations. Using the definition of the extrinsic curvature (2.5), along with (6.4) and (6.5), we obtain the following expression for the deformation action

$$\mathcal{S}_{\text{deform}} = - \kappa \int_{\Sigma_r} \sqrt{-h} \, N \left[ \mathcal{O}_{\mathcal{T}\bar{\mathcal{T}}} - \mathcal{T}_{ab} \left( \mathcal{T}_{\text{ct}}^{ab} - \frac{\mathcal{T}_{\text{ct}}}{d-1} h^{ab} \right) \right]$$

$$+ \int_{\Sigma_r} \sqrt{-h} \, U^a \, T_{sa} + \sum_{i \in \text{matter}} \int_{\Sigma_r} \sqrt{-h} \, O_i \, \partial_r J^i \, . \tag{6.9}$$

The first line contains only the gravitational contributions to the deformation, while the second line captures the matter contributions. The integrand in the first line reveals the higher-dimensional extension of the well-known $T\bar{T}$ deformation [55, 56], which takes the form

$$\mathcal{O}_{\mathcal{T}\bar{\mathcal{T}}} := \mathcal{T}^{ab} \mathcal{T}_{ab} - \frac{\mathcal{T}^2}{d-1} \, . \tag{6.10}$$

In $d = 2$, (6.10) reduces to the usual T$\bar{\text{T}}$ operator.

Next, we explore how the conjugate variables $h_{ab}$ and $\mathcal{T}^{ab}$ deform as we move in the radial direction. The key insights come from two points: 1) the BY-EMT provides the deformation flow equation for the induced metric, and 2) (6.7c) gives the deformation flow equation for the BY-EMT itself

$$
\boxed{
\begin{aligned}
\frac{1}{N}\mathcal{D}_r h_{ab} &= 2\kappa \left( \mathring{\mathcal{T}}_{ab} - \frac{\mathring{\mathcal{T}}}{d-1}\gamma_{ab} \right), \\
\frac{1}{N}\mathcal{D}_r \mathring{\mathcal{T}}_{ab} &= 2\kappa \mathring{\mathcal{T}}_a{}^c \mathring{\mathcal{T}}_{bc} - \kappa\frac{\mathring{\mathcal{T}}\mathring{\mathcal{T}}_{ab}}{d-1} + \kappa\,\mathring{\mathcal{O}}_{\mathcal{T}\bar{\mathcal{T}}} h_{ab} - \frac{1}{\kappa}\frac{\hat{\nabla}_a\hat{\nabla}_b N}{N} + \frac{1}{\kappa}\frac{\hat{\nabla}_c\hat{\nabla}^c N}{N}h_{ab} + \frac{1}{\kappa}\hat{R}_{ab} - T_{ab}.
\end{aligned}
}
\tag{6.11}
$$

Here, $\mathring{\mathcal{O}}_{\mathcal{T}\bar{\mathcal{T}}} = \mathring{\mathcal{T}}^{ab}\mathring{\mathcal{T}}_{ab} - \frac{\mathring{\mathcal{T}}^2}{d-1}$. Similar equations hold for the matter fields. These are first-order differential equations with respect to the deformation parameter $r$. In the following, we present several explicit examples of matter fields.

## 6.2 Pure GR

As a first example, we examine pure General Relativity with $\mathcal{L}_{\mathrm{M}} = 0$ and $T_{\mu\nu} = 0$. In this case, the deformation action corresponds to the first line of (6.9)

$$
d = 2: \quad \mathcal{S}_{\mathrm{deform}} = -\int_{\Sigma_c} \sqrt{-h}\, N \left[ \kappa\,\mathcal{O}_{\mathcal{T}\bar{\mathcal{T}}} + \frac{\mathcal{T}}{\ell} \right],
\tag{6.12a}
$$

$$
d = 3: \quad \mathcal{S}_{\mathrm{deform}} = -\int_{\Sigma_c} \sqrt{-h}\, N \left[ \kappa\,\mathcal{O}_{\mathcal{T}\bar{\mathcal{T}}} + \frac{\mathcal{T}}{\ell} + \ell\,\hat{R}^{ab}\,\mathcal{T}_{ab} - \frac{\ell}{4}\hat{R}\,\mathcal{T} \right],
\tag{6.12b}
$$

$$
d = 4: \quad \mathcal{S}_{\mathrm{deform}} = -\int_{\Sigma_c} \sqrt{-h}\, N \left[ \kappa\,\mathcal{O}_{\mathcal{T}\bar{\mathcal{T}}} + \frac{\mathcal{T}}{\ell} + \frac{\ell}{2}\hat{R}^{ab}\,\mathcal{T}_{ab} - \frac{\ell}{12}\hat{R}\,\mathcal{T} \right].
\tag{6.12c}
$$

These expressions illustrate the deformation action for pure GR in different spacetime dimensions. Our expressions in (6.12) are written in terms of the renormalized Brown-York energy-momentum tensor (BY-EMT), whereas those in [56] are expressed in terms of the (unrenormalized) BY-EMT. For instance, in the $d = 2$ case, when written in terms of the BY-EMT, the deformation action takes the form

$$
\mathcal{S}_{\mathrm{deform}} = -\int_{\Sigma_c} \sqrt{-h}\, N \left[ \kappa\,\mathring{\mathcal{O}}_{\mathcal{T}\bar{\mathcal{T}}} - \frac{\mathring{\mathcal{T}}}{\ell} \right].
\tag{6.13}
$$

## 6.3 Einstein-Scalar

As a second example, we consider the Einstein-Scalar theory with the following matter Lagrangian

$$
\mathcal{L}_\phi = -\frac{1}{2}\left( g^{\mu\nu}\nabla_\mu\phi\nabla_\nu\phi + m^2\phi^2 \right) - V(\phi),
\tag{6.14}
$$

where the corresponding energy-momentum tensor is given by

$$
T_{\mu\nu} = \nabla_\mu\phi\nabla_\nu\phi + \mathcal{L}_\phi\, g_{\mu\nu}.
\tag{6.15}
$$

Similar to pure General Relativity, the boundary counterterms depend on the spacetime dimension and, in this case, also on the mass of the scalar field [83]. The counterterm Lagrangian is given by

$$
\mathcal{L}_{\mathrm{ct}} = \mathcal{L}_{\mathrm{ct}}^{\mathrm{GR}} - \kappa\sqrt{-h}\left( \frac{d-\Delta}{2}\phi^2 + \frac{1}{2(2\Delta - d - 2)}\phi\Box_h\phi + \cdots \right).
\tag{6.16}
$$

Here, $\Delta$ denotes the larger root of the equation $\Delta(\Delta - d) = m^2$, the ellipsis represents higher derivative terms, and $\mathcal{L}_{\text{ct}}^{\text{GR}}$ refers to the counterterms for pure General Relativity, as defined in (6.2). In the following, we focus on the special case $\Delta = d/2 + 1$, where the coefficient of term $\phi \Box_h \phi$ is replaced by $\frac{1}{2} \ln r$ [83] and the series terminates; there is no ellipsis terms. Under this condition, the counterterm Lagrangian (6.16) truncates to

$$\mathcal{L}_{\text{ct}} = \mathcal{L}_{\text{ct}}^{\text{GR}} - \frac{\kappa}{2} \sqrt{-h} \left[ \left( \frac{d}{2} - 1 \right) \phi^2 + \ln r \, \phi \Box_h \phi \right]. \tag{6.17}$$

Now we are ready to compute the symplectic potential for this theory, which yields

$$\Theta_{\text{D}}(\Sigma_r) := \int_{\Sigma_r} \Theta_{\text{D}}^r = - \int_{\Sigma_r} \sqrt{-h} \left( \frac{1}{2} \mathcal{T}^{ab} \, \delta h_{ab} + \Pi \, \delta\phi \right) + \text{corner terms}, \tag{6.18}$$

where $\Pi$ is the normalized canonical conjugate of the scalar field $\phi$

$$\Pi = \mathring{\Pi} + \Pi_{\text{ct}}, \qquad \mathring{\Pi} = N^{-1} \mathcal{D}_r \phi, \qquad \Pi_{\text{ct}} = \left( \frac{d}{2} - 1 \right) \phi + \ln r \, \Box_h \phi. \tag{6.19}$$

Using this setup, we can compute the deformation action

$$\mathcal{S}_{\text{deform}} = - \int_{\Sigma_c} \sqrt{-h} \, N \left[ \kappa \, \mathcal{O}_{\mathcal{T}\bar{\mathcal{T}}} - \kappa \mathcal{T}_{ab} \left( \mathcal{T}_{\text{ct}}^{ab} - \frac{\mathcal{T}_{\text{ct}}}{d-1} h^{ab} \right) + \Pi^2 - \Pi_{\text{ct}} (\Pi - N^{-1} U^a \partial_a \phi) \right]. \tag{6.20}$$

This expression defines the deformation action for the Einstein-scalar theory, incorporating both gravitational and matter contributions. Finally, we note that in the special case where gravitational contributions are turned off and $U^a = 0$, our result reduces to that of [56].

## 6.4 Einstein-Maxwell

As the third example, we study the Einstein-Maxwell theory, where the electromagnetic Lagrangian is given by

$$\mathcal{L}_{\text{EM}} = -\frac{1}{4} F_{\mu\nu} F^{\mu\nu}, \tag{6.21}$$

with the associated Maxwell energy-momentum tensor given by

$$T_{\mu\nu} = F_{\mu\alpha} F_\nu{}^\alpha + \mathcal{L}_{\text{EM}} \, g_{\mu\nu}. \tag{6.22}$$

The symplectic form of this theory is given by

$$\Theta_{\text{D}}(\Sigma_r) = - \int_{\Sigma_r} \sqrt{-h} \left( \frac{1}{2} \mathcal{T}^{ab} \, \delta h_{ab} + \Pi^a \, \delta A_a \right) + \text{corner terms}, \tag{6.23}$$

where $\Pi^a = N^{-1}(F_r{}^a - U^b F_b{}^a)$ is the canonical conjugate of the gauge field $A_a$. Finally, the deformation action is

$$\mathcal{S}_{\text{deform}} = - \int_{\Sigma_c} \sqrt{-h} \, N \left[ \kappa \, \mathcal{O}_{\mathcal{T}\bar{\mathcal{T}}} - \kappa \mathcal{T}_{ab} \left( \mathcal{T}_{\text{ct}}^{ab} - \frac{\mathcal{T}_{\text{ct}}}{d-1} h^{ab} \right) + \left( \Pi_a + \frac{U^b}{2N} F_{ba} \right) \Pi^a \right]. \tag{6.24}$$

This gives the deformation action for the Einstein-Maxwell theory, including both the gravitational and gauge field contributions.

## 6.5 Einstein-Chern-Simon

As the next example, we consider the Abelian Einstein-Chern-Simons theory in $d = 2n$, with the following Lagrangian

$$\mathcal{L}_{\text{ACS}} = \gamma \, \epsilon^{\rho \mu_1 \nu_1 \cdots \mu_n \nu_n} A_\rho F_{\mu_1 \nu_1} \cdots F_{\mu_n \nu_n} \,, \tag{6.25}$$

where $\gamma$ is the coupling constant. The Chern-Simons term is a topological contribution, meaning it is independent of the background metric. Consequently, its energy-momentum tensor vanishes, i.e., $T_{\mu\nu} = 0$. Furthermore, the equation of motion enforces

$$\epsilon^{\rho \mu_1 \nu_1 \cdots \mu_n \nu_n} F_{\mu_1 \nu_1} \cdots F_{\mu_n \nu_n} = 0 \,. \tag{6.26}$$

As a result, the on-shell matter Lagrangian also vanishes, $\mathcal{L}_{\text{ACS}} = 0$, leaving the deformation action with only the gravitational contribution, as given by the first line of (6.9) for $d = 2n$.

## 6.6 Radial deformation in Lovelock theories

As the final example in this section, we examine radial deformations in Lovelock theories. Lovelock theories have second-order equations of motion, and the corresponding symplectic potential, consistent with Dirichlet boundary conditions, takes the form

$$\Theta_{\text{D}}(\Sigma_r) = -\frac{1}{2} \int_{\Sigma_r} \sqrt{-h} \, \mathcal{T}^{ab} \, \delta h_{ab} + \text{corner terms} \,, \tag{6.27}$$

where $\mathcal{T}^{ab}$ denotes the rBY-EMT for Lovelock theories. From (3.32) (neglecting corner terms as before), we obtain the deformation action

$$\mathcal{S}_{\text{deform}} = -\frac{1}{2} \int_{\Sigma_r} \sqrt{-h} \, \mathcal{T}^{ab} \, \partial_r h_{ab} = -\int_{\Sigma_r} \sqrt{-h} \, N \, K_{ab} \, \mathcal{T}^{ab} \,. \tag{6.28}$$

In the second equality, we have used $\hat{\nabla}_a \mathcal{T}^{ab} = 0$ and have once again omitted corner terms. To proceed, we express $K_{ab}$ in terms of $\mathcal{T}^{ab}$, which necessitates specifying the explicit form of the theory. As an illustrative example in this theory, we consider the Einstein-Gauss-Bonnet Lagrangian

$$\mathcal{L}_{\text{bulk}}^{\text{D}} = \frac{\sqrt{-g}}{2\kappa} \left[ R - 2\Lambda + \alpha \left( R^2 - 4R_{\mu\nu} R^{\mu\nu} + R^{\mu\nu\alpha\beta} R_{\mu\nu\alpha\beta} \right) \right] + \frac{1}{\kappa} (\mathcal{L}_{\text{GHY}} + \mathcal{L}_{\text{ct}}) \,, \tag{6.29}$$

where the Gibbons-Hawking-York term, which is the boundary term guaranteeing the Dirichlet boundary conditions, is given by [84, 85]

$$\mathcal{L}_{\text{GHY}} = \sqrt{-h} \left[ K - \frac{2\alpha}{3} \left( K^3 - 3KK^{(2)} + 2K^{(3)} \right) - 4\alpha \hat{G}_{ab} K^{ab} \right] \,, \tag{6.30}$$

where $K^{(2)} \equiv K_b^a K_a^b$ and $K^{(3)} \equiv K_b^a K_c^b K_a^c$. The variation of the total action leads to [84]

$$G_{\mu\nu} + \Lambda g_{\mu\nu} + 2\alpha \, H_{\mu\nu} = 0 \,, \tag{6.31}$$

where $H_{\mu\nu}$ captures the contribution of the Gauss-Bonnet Lagrangian

$$H_{\mu\nu} = R R_{\mu\nu} - 2R_{\mu\alpha} R^\alpha{}_\nu - 2R^{\alpha\beta} R_{\mu\alpha\nu\beta} + R_\mu{}^{\alpha\beta\delta} R_{\nu\alpha\beta\delta} - \frac{1}{4} g_{\mu\nu} \left( R^2 - 4R_{\alpha\beta} R^{\alpha\beta} + R^{\alpha\beta\delta\tau} R_{\alpha\beta\delta\tau} \right) \,. \tag{6.32}$$

For this theory, the rBY-EMT is given by

$$\mathcal{T}_{ab} = \mathring{\mathcal{T}}_{ab} + \mathcal{T}_{ab}^{\text{ct}} \,, \qquad \mathring{\mathcal{T}}_{ab} = \mathring{\mathcal{T}}_{ab}^{\text{EH}} + \alpha \mathring{\mathcal{T}}_{ab}^{\text{GB}} \,, \tag{6.33}$$

where $\mathring{\mathcal{T}}_{ab}^{\text{EH}}$ and $\mathring{\mathcal{T}}_{\text{GB}}^{ab}$ are as follows

$$\mathring{\mathcal{T}}_{ab}^{\text{EH}} = \frac{1}{\kappa}\left(K_{ab} - K\,h_{ab}\right), \tag{6.34}$$

$$\mathring{\mathcal{T}}_{ab}^{\text{GB}} = \frac{2}{\kappa}\Big[\left(\hat{R} + K^{(2)} - K^2\right)K_{ab} + 2\left(K\,\hat{G}_{ab} - 2K_{(a}^c\hat{R}_{b)c} + KK_a^cK_{bc} - K_a^cK_{cd}K_b^d - \hat{R}_{acbd}K^{cd}\right)$$
$$+ \left(\frac{1}{3}K^3 - KK^{(2)} + \frac{2}{3}K^{(3)} + 2\hat{R}_{ab}K^{ab}\right)h_{ab}\Big]. \tag{6.35}$$

The dynamical equations are

$$\hat{R} - 2\Lambda + \kappa\,K_{ab}\mathring{\mathcal{T}}^{ab} + \alpha\Big(\frac{K^4}{3} - 2K^2K^{(2)} + (K^{(2)})^2 - 2K^{(4)} + \frac{8}{3}K\,K^{(3)}$$
$$+ \hat{R}^2 - 4\hat{R}_{ab}\hat{R}^{ab} + \hat{R}^{abcd}\hat{R}_{abcd}\Big) = 0, \tag{6.36a}$$

$$\hat{\nabla}_a\mathring{\mathcal{T}}^{ab} = 0, \tag{6.36b}$$

$$\partial_r\mathring{\mathcal{T}}_{ab} + \cdots = 0, \tag{6.36c}$$

where $K^{(4)} \equiv K_b^aK_c^bK_e^cK_a^e$. The above equations are written in a specific gauge, where $N = 1$ and $U^a = 0$. The ellipses in (6.36c) denote a lengthy expression that is not crucial for the subsequent analysis, so we omit its explicit form. As mentioned earlier, the explicit form of the counterterms depends on the spacetime dimension. In particular, the Gauss-Bonnet term in the Lagrangian is purely topological for $d < 4$ and does not contribute to the equations of motion. For instance, in $d = 4$, the counterterm Lagrangian is given by [85]

$$\mathcal{L}_{\text{ct}} = -\sqrt{-h}\left[\frac{3}{\ell}\left(1 + \frac{\ell^2}{12}\hat{R}\right) + \frac{\alpha}{\ell}\left(-\frac{1}{\ell^2} + \frac{3}{4}\hat{R}\right)\right], \tag{6.37}$$

where $\hat{R}$ is the Ricci scalar associated with the induced metric $h_{ab}$. The corresponding counter energy-momentum tensor is given by

$$\kappa\,\mathcal{T}_{\text{ct}}^{ab} = \frac{1}{\ell}\left[3h^{ab} - \frac{\ell^2}{2}\hat{G}^{ab} - \frac{\alpha}{\ell^2}\left(h^{ab} + \frac{3}{2}\ell^2\hat{G}^{ab}\right)\right]. \tag{6.38}$$

It is important to emphasize that the BY-EMT in (6.33)-(6.35) is expressed in terms of the extrinsic curvature. In principle, this equation can be inverted to express the extrinsic curvature as a function of the BY-EMT and the induced metric, i.e., $K_{ab} = K_{ab}[\mathring{\mathcal{T}}_{cd}; h_{cd}]$. Consequently, the right-hand side of (6.28) can be interpreted as a function of the BY-EMT. However, as is evident from its form, the extrinsic curvature is a highly nontrivial function of the BY-EMT. Here, we determine the explicit form of $\mathcal{S}_{\text{deform}}$ up to linear order in $\alpha$

$$\mathcal{S}_{\text{deform}} = -\int_{\Sigma_r}\sqrt{-h}\,N\bigg\{\mathring{\mathcal{O}}_{\mathcal{T}\bar{\mathcal{T}}} + 2\alpha\Big[\mathring{\mathcal{T}}_{ab}^{\text{GB}}\mathring{\mathcal{T}}_{\text{EH}}^{ab} - \frac{2d-1}{2d(d-1)}\mathring{\mathcal{T}}_{\text{EH}}\mathring{\mathcal{T}}_{\text{GB}} - \hat{R}\,\mathring{\mathcal{T}}_{\text{EH}}^{(2)} + \frac{d+3}{d(d-1)}\hat{R}\,\mathring{\mathcal{T}}_{\text{EH}}^2$$
$$+ 4\hat{R}_b^a\,\mathring{\mathcal{T}}_{ac}^{\text{EH}}\mathring{\mathcal{T}}_{\text{EH}}^{bc} - \frac{2(5d-3)}{d(d-1)}\mathring{\mathcal{T}}_{\text{EH}}\,\hat{R}^{ab}\mathring{\mathcal{T}}_{ab}^{\text{EH}} + 2\hat{R}_{acbd}\,\mathring{\mathcal{T}}_{\text{EH}}^{ab}\mathring{\mathcal{T}}_{\text{EH}}^{cd} + \mathring{\mathcal{T}}_{\text{EH}}^{(4)} - \mathring{\mathcal{T}}_{\text{EH}}^{(2)}\mathring{\mathcal{T}}_{\text{EH}}^{(2)}$$
$$- \frac{d+1}{d(d-1)^2}\mathring{\mathcal{T}}_{\text{EH}}^4 + \frac{(2d^2+3d-3)}{d(d-1)^2}\mathring{\mathcal{T}}_{\text{EH}}^2\,\mathring{\mathcal{T}}_{\text{EH}}^{(2)} - \frac{2(3d-1)}{d(d-1)}\mathring{\mathcal{T}}_{\text{EH}}\mathring{\mathcal{T}}_{\text{EH}}^{(3)}\Big] + \mathcal{O}(\alpha^2)\bigg\}, \tag{6.39}$$

where

$$\mathring{\mathcal{O}}_{\mathcal{T}\bar{\mathcal{T}}} = \mathring{\mathcal{T}}_{\text{EH}}^{ab}\mathring{\mathcal{T}}_{ab}^{\text{EH}} - \frac{\mathring{\mathcal{T}}_{\text{EH}}^2}{d-1}, \qquad \mathring{\mathcal{T}}_{\text{EH}}^{(2)} \equiv (\mathring{\mathcal{T}}_{\text{EH}})_b^a(\mathring{\mathcal{T}}_{\text{EH}})_a^b,$$
$$\mathring{\mathcal{T}}_{\text{EH}}^{(3)} \equiv (\mathring{\mathcal{T}}_{\text{EH}})_b^a(\mathring{\mathcal{T}}_{\text{EH}})_c^b(\mathring{\mathcal{T}}_{\text{EH}})_a^c, \qquad \mathring{\mathcal{T}}_{\text{EH}}^{(4)} \equiv (\mathring{\mathcal{T}}_{\text{EH}})_b^a(\mathring{\mathcal{T}}_{\text{EH}})_c^b(\mathring{\mathcal{T}}_{\text{EH}})_e^c(\mathring{\mathcal{T}}_{\text{EH}})_a^e. \tag{6.40}$$

We expressed $\mathcal{S}_{\text{deform}}$ in terms of the BY-EMT, however, upon (6.33) it can also be rewritten in terms of the rBY-EMT.

# 7   Radial evolution of arbitrary boundary conditions

In Section 5.2, we constructed the boundary deformation procedure that facilitates the transition from boundary condition X on $\Sigma_r$ to boundary condition Y on $\Sigma_{r+\mathrm{d}r}$. In this section, we present concrete examples of such transitions. Specifically, we consider the following cases:

$$X \to Y, \qquad \text{where} \qquad X, Y \in \{\text{Dirichlet, Neumann, Conformal, Conformal Conjugate}\}. \qquad (7.1)$$

For simplicity, we focus on pure gravity without matter fields; however, incorporating matter fields is a straightforward extension of the examples from the previous section. Throughout this section, we ignore corner terms and consider only codimension-one contributions. We should note that our definition of Neumann and conformal boundary conditions are as defined in [34] (cf. section 5 there), which is slightly different than the one used in the literature.

## 7.1   Dirichlet to different boundary conditions

In this subsection, we examine the transition from Dirichlet boundary conditions at radius $r_1$ to Dirichlet, Neumann, Conformal, and Conformal Conjugate boundary conditions at radius $r_2$.

**Dirichlet to Dirichlet.**   The generator that governs the transition between different radii with distinct boundary conditions is given by (5.11)

$$\mathcal{S}_{X \to Y} = \mathrm{d}r \int_{\Sigma_r} (\mathcal{L}_{\text{bulk}}^{\text{D}} + \partial_r W_{\text{bulk}}) + \int_{\Sigma_r} W_{X \to Y}. \qquad (7.2)$$

The Dirichlet boundary condition on $\Sigma_r$ is defined as $\delta h_{ab}\big|_{\Sigma_r} = 0$. For the Dirichlet-to-Dirichlet transition $(X \equiv D \to Y \equiv D)$, we set

$$W_{\text{bulk}} = 0, \qquad W_{D \to D} = 0. \qquad (7.3)$$

As the result, we obtain (6.12), where

$$\mathcal{S}_{D \to D} = \mathrm{d}r \int_{\Sigma_r} \mathcal{L}_{\text{bulk}}^{\text{D}} = \mathrm{d}r \, \mathcal{S}_{\text{deform}}. \qquad (7.4)$$

This type of transition corresponds to the seminal $\text{T}\bar{\text{T}}$ deformation.

**Dirichlet to Neumann.**   The Neumann boundary condition on $\Sigma_r$ is defined as $\delta(\sqrt{-h}\,\mathcal{T}^{ab})\big|_{\Sigma_r} = 0$. For Dirichlet to Neumann transition, we choose

$$W_{\text{bulk}} = 0, \qquad W_{D \to N} = \frac{1}{2}\sqrt{-h}\,\mathcal{T}. \qquad (7.5)$$

Therefore, the generator of the Dirichlet to Neumann transition is given by

$$\mathcal{S}_{D \to N} = \mathcal{S}_{D \to D} + \frac{1}{2}\int_{\Sigma_r} \sqrt{-h}\,\mathcal{T}. \qquad (7.6)$$

**Dirichlet to Conformal.** The conformal boundary condition [59, 62–64, 86–90] on $\Sigma_r$ is defined by $\delta g_{ab}|_{\Sigma_r} = 0$ and $\delta\mathcal{T}|_{\Sigma_r} = 0$, where the determinant-free boundary metric is given by $g_{ab} := (-h)^{-1/d}h_{ab}$. The Dirichlet-to-conformal transition provides

$$W_{\text{bulk}} = 0, \qquad W_{D\to C} = \frac{1}{d}\sqrt{-h}\,\mathcal{T}. \tag{7.7}$$

Hence,

$$\mathcal{S}_{D\to C} = \mathcal{S}_{D\to D} + \frac{1}{d}\int_{\Sigma_r}\sqrt{-h}\,\mathcal{T}. \tag{7.8}$$

**Dirichlet to Conformal Conjugate.** The conformal conjugate boundary conditions were first introduced in [34]. These conditions are defined as $\delta(\sqrt{-h}\,\mathrm{T}^{ab})|_{\Sigma_r} = 0$ and $\delta\sqrt{-h}|_{\Sigma_r} = 0$, where $\mathrm{T}^{ab}$ is the rescaled, trace-free EMT

$$\mathrm{T}^{ab} := (-h)^{1/d}\left(\mathcal{T}^{ab} - \frac{\mathcal{T}}{d}h^{ab}\right).$$

This transition from Dirichlet to conformal conjugate occurs under the conditions

$$W_{\text{bulk}} = 0, \qquad W_{D\to CC} = 0, \tag{7.9}$$

which match those of the Dirichlet case (7.3). Consequently, the deformation remains identical to (7.4).

## 7.2 Neumann to different boundary conditions

Here in this subsection, we study the transition from Neumann boundary conditions at radius $r_1$ to Neumann, Dirichlet, Conformal, and Conformal Conjugate boundary conditions at radius $r_2$.

**Neumann to Neumann.** We are now considering the transition from Neumann to Neumann boundary conditions. To implement this transition, we choose $W_{\text{bulk}}$ and $W_{N\to N}$ as follows

$$W_{\text{bulk}} = \frac{1}{2}\sqrt{-h}\,\mathcal{T}, \qquad W_{N\to N} = 0. \tag{7.10}$$

With this choice, the deformation action takes the form

$$\mathcal{S}_{N\to N} = \mathrm{d}r\int_{\Sigma_r}\left(\mathcal{L}_{\text{bulk}}^{\mathrm{D}} + \partial_r W_{\text{bulk}}\right). \tag{7.11}$$

To complete the derivation, we express $\partial_r W_{\text{bulk}}$ in terms of boundary variables. The calculations are straightforward; for details, see Appendix A. The boundary deformation action for different dimensions is given by

$$\begin{aligned}
d=2: \quad &\mathcal{S}_{N\to N} = \mathrm{d}r\int_{\Sigma_r} N\sqrt{-h}\left[-\frac{\kappa}{2}\mathcal{O}_{\mathcal{T}\bar{\mathcal{T}}} - \frac{\mathcal{T}}{\ell}\right], \\
d=3: \quad &\mathcal{S}_{N\to N} = \mathrm{d}r\int_{\Sigma_r} N\sqrt{-h}\left[-\frac{\mathcal{T}}{2\ell} + \frac{\ell}{2}\hat{R}_{ab}\mathcal{T}^{ab} - \frac{\ell}{8}\hat{R}\mathcal{T} + \frac{\ell^2}{2}\hat{R}_{ab}\hat{R}^{ab} - \frac{3\ell^2}{16}\hat{R}^2\right], \\
d=4: \quad &\mathcal{S}_{N\to N} = \mathrm{d}r\int_{\Sigma_r} N\sqrt{-h}\left[\frac{\kappa}{2}\mathcal{O}_{\mathcal{T}\bar{\mathcal{T}}} + \frac{\ell}{2}\hat{R}_{ab}\mathcal{T}^{ab} + \frac{\ell}{4}\hat{R}\mathcal{T} + \frac{\ell^2}{8}\hat{R}_{ab}\hat{R}^{ab} - \frac{\ell^2}{24}\hat{R}^2\right].
\end{aligned} \tag{7.12}$$

As seen from the second line of the above equation, the $\mathrm{T}\bar{\mathrm{T}}$ operator does not appear in asymptotic $\text{AdS}_4$ bulk spacetime. This follows directly from the fact that in four dimensions, a well-defined action principle with Neumann boundary conditions does not require the Gibbons-Hawking-York term [91]. More precisely, the coefficient of $\mathcal{O}_{\mathcal{T}\bar{\mathcal{T}}}$ in $d+1$ spacetime dimensions is given by $\frac{d-3}{2}$, which is negative for $d=2$, zero for $d=3$, and positive for all $d>3$.

**Neumann to Dirichlet.** The transition from Neumann to Dirichlet boundary conditions is implemented through the following choices

$$W_{\text{bulk}} = \frac{1}{2}\sqrt{-h}\,\mathcal{T}\,, \qquad W_{N\to D} = -\frac{1}{2}\sqrt{-h}\,\mathcal{T}\,. \tag{7.13}$$

As a result, the boundary deformation action takes the form

$$\mathcal{S}_{N\to D} = \mathcal{S}_{N\to N} - \frac{1}{2}\int_{\Sigma_r}\sqrt{-h}\,\mathcal{T}\,. \tag{7.14}$$

**Neumann to Conformal.** For Neumann to Conformal transition, we choose

$$W_{\text{bulk}} = \frac{1}{2}\sqrt{-h}\,\mathcal{T}\,, \qquad W_{N\to C} = -\frac{d-2}{2d}\sqrt{-h}\,\mathcal{T}\,. \tag{7.15}$$

Hence,

$$\mathcal{S}_{N\to C} = \mathcal{S}_{N\to N} - \frac{d-2}{2d}\int_{\Sigma_r}\sqrt{-h}\,\mathcal{T}. \tag{7.16}$$

**Neumann to Conformal Conjugate.** For this transition, the $W$ terms must be chosen as in (7.13). Consequently, the deformation action is given by (7.14).

## 7.3 Conformal to different boundary conditions

For the next case, we examine the transition from Conformal boundary conditions at radius $r_1$ to Conformal, Dirichlet, Neumann, and Conformal Conjugate boundary conditions at radius $r_2$.

**Conformal to Conformal.** We now consider the transition from Conformal to Conformal boundary conditions. To implement this transition, we choose

$$W_{\text{bulk}} = \frac{1}{d}\sqrt{-h}\,\mathcal{T}\,, \qquad W_{C\to C} = 0\,. \tag{7.17}$$

This leads to the following boundary deformation action:

$$
\begin{aligned}
d = 2: \quad &\mathcal{S}_{C\to C} = \mathrm{d}r\int_{\Sigma_r} N\sqrt{-h}\left[-\frac{\kappa}{2}\mathcal{O}_{\mathcal{T}\bar{\mathcal{T}}} - \frac{\mathcal{T}}{\ell}\right], \\
d = 3: \quad &\mathcal{S}_{C\to C} = \mathrm{d}r\int_{\Sigma_r} N\sqrt{-h}\left[-\frac{\kappa}{3}\mathcal{O}_{\mathcal{T}\bar{\mathcal{T}}} - \frac{2\mathcal{T}}{\ell} + \frac{\ell^2}{3}\hat{R}_{ab}\hat{R}^{ab} - \frac{\ell^2}{8}\hat{R}^2\right], \\
d = 4: \quad &\mathcal{S}_{C\to C} = \mathrm{d}r\int_{\Sigma_r} N\sqrt{-h}\left[-\frac{\kappa}{4}\mathcal{O}_{\mathcal{T}\bar{\mathcal{T}}} + \hat{R}_{ab}\mathcal{T}^{ab} - \frac{\mathcal{T}}{2\ell} + \frac{\ell^2}{16}\hat{R}_{ab}\hat{R}^{ab} - \frac{\ell^2}{48}\hat{R}^2\right].
\end{aligned}
\tag{7.18}
$$

The coefficient of $\mathcal{O}_{\mathcal{T}\bar{\mathcal{T}}}$ in $d+1$ spacetime dimensions is $-\frac{1}{d}$, which remains negative in all dimensions.

**Conformal to Dirichlet.** To transition from Conformal to Dirichlet, we choose the following $W$-terms

$$W_{\text{bulk}} = \frac{1}{d}\sqrt{-h}\,\mathcal{T}\,, \qquad W_{C\to D} = -\frac{1}{d}\sqrt{-h}\,\mathcal{T}\,. \tag{7.19}$$

Thus, the boundary deformation action takes the form

$$\mathcal{S}_{C\to D} = \mathcal{S}_{C\to C} - \frac{1}{d}\int_{\Sigma_r}\sqrt{-h}\,\mathcal{T}\,. \tag{7.20}$$

**Conformal to Neumann.** To transition from Conformal to Neumann, we choose the following $W$-terms

$$W_{\text{bulk}} = \frac{1}{d}\sqrt{-h}\,\mathcal{T}\,, \qquad W_{C\to N} = \frac{d-2}{2d}\sqrt{-h}\,\mathcal{T}\,. \tag{7.21}$$

As a result, the boundary deformation action becomes

$$\mathcal{S}_{C\to N} = \mathcal{S}_{C\to C} + \frac{d-2}{2d}\int_{\Sigma_r}\sqrt{-h}\,\mathcal{T}\,. \tag{7.22}$$

**Conformal to Conformal Conjugate.** This transition occurs with $W$ terms (7.19). Consequently, the boundary deformation remains the same as in (7.20).

We conclude this section by noting that the transition from Conformal Conjugate boundary conditions to other boundary conditions follows the same procedure as in the Dirichlet case (see Section 7.1). Therefore, we do not repeat it here. To clarify this point further, we recall that in [34], we demonstrated that a given $W$-term corresponds to two distinct boundary conditions. While that analysis focused on asymptotic infinity, the result is general and holds at any finite radius as well. By construction, these two boundary conditions share the same $W$-terms and, consequently, the same boundary deformation action.

## 8 Hydrodynamic deformations

In this section, we investigate hydrodynamic deformations—transformations that map one hydrodynamic system to another. We start with a phase space described by the canonical pair $(h_{ab}, \sqrt{-h}\,\mathcal{T}^{ab})$ and, after the deformation, obtain a new canonical pair $(\tilde{h}_{ab}, \sqrt{-\tilde{h}}\,\tilde{\mathcal{T}}^{ab})$. Both configurations preserve their hydrodynamic interpretation, satisfying the conservation laws $\hat{\nabla}_a\mathcal{T}^{ab} = 0$ and $\tilde{\nabla}_a\tilde{\mathcal{T}}^{ab} = 0$, where $\hat{\nabla}_a$ and $\tilde{\nabla}_a$ denote the covariant derivatives associated with $h_{ab}$ and $\tilde{h}_{ab}$, respectively.

We focus on two distinct classes of hydrodynamic deformations: 1. Extrinsic hydrodynamic deformation: A one-function family of deformations characterized by an arbitrary function of the trace of the EMT. 2. Intrinsic hydrodynamic deformation: A class of deformations defined in terms of an arbitrary function of the intrinsic geometric variables of the boundary. The first class of deformations was initially introduced in [92], with a special case examined in [63]. It was later explored in detail within the framework of freelance holography [34]. The second class is often employed as a regularization procedure for boundary variables.

### 8.1 Extrinsic hydrodynamic deformation

Consider the following one-function family of extrinsic deformations:

$$\tilde{h}_{ab} = \Omega^{-1}h_{ab}\,, \qquad \tilde{\mathcal{T}}^{ab} = \Omega^{1+\frac{d}{2}}\left[\mathcal{T}^{ab} - \frac{1}{2}G(\mathcal{T})h^{ab}\right]\,, \tag{8.1}$$

where $\Omega = \Omega(\mathcal{T})$ is an arbitrary function of the trace of the rBY-EMT. This function characterizes the one-function family of hydrodynamic deformations. The function $G(\mathcal{T})$ is determined in terms of $\Omega$ as

$$G(\mathcal{T}) = \frac{2}{d}\left(\mathcal{T} - \Omega^{-d/2}\int_0^{\mathcal{T}}\Omega^{d/2}\,d\mathcal{T}\right)\,. \tag{8.2}$$

The Dirichlet symplectic potentials of the two hydrodynamic frames are related as

$$-\frac{1}{2}\int_{\Sigma_r}\sqrt{-\tilde{h}}\,\tilde{\mathcal{T}}^{ab}\,\delta\tilde{h}_{ab} = -\frac{1}{2}\int_{\Sigma_r}\sqrt{-h}\,\mathcal{T}^{ab}\,\delta h_{ab} + \frac{1}{2}\delta\left(\int_{\Sigma_r}\sqrt{-h}\,G(\mathcal{T})\right)\,. \tag{8.3}$$

This relation encapsulates how the symplectic structure transforms under the hydrodynamic deformations.

We would like to emphasize here a key point. In this paper, we introduced various deformations, with a particular focus on $T\bar{T}$-like deformations, which involve quadratic combinations of the EMT. Additionally, we considered another class of deformations—those governed by an arbitrary function of the trace of the EMT—which we explored in detail within the framework of freelance holography [34]. The crucial insight we wish to highlight is that Einstein's equations establish a connection between these different types of deformations. To make this connection explicit, let us examine (6.7a)

$$\hat{R} - 2\Lambda + \kappa^2 \mathcal{O}_{\mathcal{T}\bar{\mathcal{T}}} + \kappa^2 \mathcal{O}_{\mathcal{T}\bar{\mathcal{T}}}^{\text{ct}} - 2\kappa^2 \left( \mathcal{T}^{ab} \mathcal{T}_{ab}^{\text{ct}} + \frac{\mathcal{T}\mathcal{T}_{\text{ct}}}{d-1} \right) = 0 \,, \tag{8.4}$$

where we have omitted matter fields and used the identity

$$\mathring{\mathcal{O}}_{\mathcal{T}\bar{\mathcal{T}}} = \mathcal{O}_{\mathcal{T}\bar{\mathcal{T}}} + \mathcal{O}_{\mathcal{T}\bar{\mathcal{T}}}^{\text{ct}} - 2 \left( \mathcal{T}^{ab} \mathcal{T}_{ab}^{\text{ct}} - \frac{\mathcal{T}\mathcal{T}_{\text{ct}}}{d-1} \right) \,,$$

where $\mathcal{O}_{\mathcal{T}\bar{\mathcal{T}}}^{\text{ct}} := \mathcal{T}_{\text{ct}}^{ab} \mathcal{T}_{ab}^{\text{ct}} - \frac{\mathcal{T}_{\text{ct}}^2}{d-1}$. Equation (8.4) establishes a relationship between linear and quadratic combinations of the rBY-EMT. In the special case of $d = 2$, this relation simplifies to

$$\kappa^2 \mathcal{O}_{\mathcal{T}\bar{\mathcal{T}}} + \frac{2\kappa}{\ell} \mathcal{T} + \hat{R} = 0 \,, \tag{8.5}$$

where we have used (6.6). For the higher-dimensional cases, see (A.7). This result implies that trace deformations—governed by an arbitrary function of $\mathcal{T}$—can alternatively be interpreted as deformations involving an arbitrary function of $\mathcal{O}_{\mathcal{T}\bar{\mathcal{T}}}$, albeit with an additional curvature term $\hat{R}$.

## 8.2 Intrinsic hydrodynamic deformations

In this subsection, we investigate another class of hydrodynamic deformations. We begin with the following deformation of the Dirichlet symplectic potential, given by

$$-\frac{1}{2} \int_{\Sigma_r} \sqrt{-h} \, \mathcal{T}^{ab} \, \delta h_{ab} + \delta \int_{\Sigma_r} \sqrt{-h} \, W_{\text{int}}[h_{ab}, \hat{R}_{abcd}, \hat{\nabla}] \,, \tag{8.6}$$

where $W_{\text{int}}[h_{ab}, \hat{R}_{abcd}, \hat{\nabla}]$ is constructed purely from intrinsic boundary variables. Recall that

$$\delta(\sqrt{-h} \, W_{\text{int}}[h_{ab}, \hat{R}_{abcd}, \hat{\nabla}]) = -\frac{1}{2} \sqrt{-h} \, \mathcal{T}_{\text{int}}^{ab} \delta h_{ab} + \text{corner terms} \,, \tag{8.7}$$

and that the Bianchi identity associated with $W_{\text{int}}[h_{ab}, \hat{R}_{abcd}, \hat{\nabla}]$ guarantees the conservation of $\mathcal{T}_{\text{int}}^{ab}$, i.e.,

$$\hat{\nabla}_a \mathcal{T}_{\text{int}}^{ab} = 0 \,. \tag{8.8}$$

Combining (8.7) with (8.6), we obtain

$$-\frac{1}{2} \int_{\Sigma_r} \sqrt{-h} \, \mathcal{T}^{ab} \, \delta h_{ab} + \delta \int_{\Sigma_r} \sqrt{-h} \, W_{\text{int}}[h_{ab}, \hat{R}_{abcd}, \hat{\nabla}] = -\frac{1}{2} \int_{\Sigma_r} \sqrt{-h} \, \mathrm{T}^{ab} \, \delta h_{ab} \,, \tag{8.9}$$

where we define the modified stress tensor as

$$\mathrm{T}^{ab} := \mathcal{T}^{ab} + \mathcal{T}_{\text{int}}^{ab} \,, \qquad \hat{\nabla}_a \mathrm{T}^{ab} = 0 \,. \tag{8.10}$$

From (8.9) and (8.10), it follows that the canonical pair $(h_{ab}, \sqrt{-h} \, \mathrm{T}^{ab})$ retains a hydrodynamic interpretation, still with Dirichlet boundary conditions $\delta h_{ab} = 0$ on $\Sigma_r$.

## 8.3 Evolving radially among hydrodynamic frames

In the previous sections, we introduced radial evolving deformations ($T\bar{T}$-like deformations) for transitions between boundaries at different radii and with different boundary conditions. Now, we address the question of whether these deformations are hydrodynamic in nature. Recall that we provided two distinct interpretations for $T\bar{T}$-like deformations. In the radial evolving picture, the answer to this question is clearly affirmative. For example, in the Dirichlet-to-Dirichlet transition, we have a hydrodynamic frame on $\Sigma_r$ with $(h_{ab}, \sqrt{-h}\,\mathcal{T}^{ab})\big|_{\Sigma_r}$ and evolve to another hydrodynamic frame on $\Sigma_{r+\mathrm{d}r}$ with $(h_{ab}, \sqrt{-h}\,\mathcal{T}^{ab})\big|_{\Sigma_{r+\mathrm{d}r}}$. Next, we ask whether the second interpretation also describes a hydrodynamic system. To explore this, we present Dirichlet-to-Dirichlet boundary deformations in $d = 2$, with more general cases following as extensions along similar lines.

In $d = 2$, the deformation action is given by (6.12a)

$$\mathcal{S}_{\text{deform}} = - \int_{\Sigma_r} \sqrt{-h}\, N \left[ \kappa\, \mathcal{O}_{\mathcal{T}\bar{\mathcal{T}}} + \ell^{-1}\, \mathcal{T} \right]. \tag{8.11}$$

Using (8.5), we now express $\mathcal{O}_{\mathcal{T}\bar{\mathcal{T}}}$ in terms of $\hat{R}$ and $\mathcal{T}$. With this relation, we can rewrite the deformation action as

$$\mathcal{S}_{\text{deform}} = \int_{\Sigma_r} \sqrt{-h} \left( \kappa^{-1}\, \hat{R} + \frac{\mathcal{T}}{\ell} \right), \tag{8.12}$$

where we set $N = 1$. Varying $\mathcal{S}_{\text{deform}}$ gives

$$\delta \mathcal{S}_{\text{deform}} = -\kappa^{-1} \int_{\Sigma_r} \sqrt{-h}\, \hat{G}^{ab}\, \delta h_{ab} + \delta \int_{\Sigma_r} \sqrt{-h}\, \ell^{-1}\, \mathcal{T}. \tag{8.13}$$

Rewriting (5.5) for the special case we are exploring here, we obtain

$$-\frac{1}{2} \int_{\Sigma_r} \sqrt{-h}\, \mathcal{T}^{ab}\, \delta h_{ab} + \mathrm{d}r\, \delta \mathcal{S}_{\text{deform}} = -\frac{1}{2} \int_{\Sigma_r} \sqrt{-h}\, \mathrm{T}^{ab}\, \delta h_{ab} + \mathrm{d}r\, \delta \int_{\Sigma_r} \sqrt{-h}\, \ell^{-1}\, \mathrm{T}, \tag{8.14}$$

where $\mathrm{T}^{ab}$ is given by

$$\mathrm{T}^{ab} = \mathcal{T}^{ab} + 2\, \mathrm{d}r\, \kappa^{-1} \hat{G}^{ab}, \qquad \mathrm{T} = \mathcal{T}. \tag{8.15}$$

(Note that in $d = 2$ Einstein tensor $\hat{G}^{ab}$ is traceless.) Our final step is to absorb the last term in (8.14) into the canonical term. Comparing with (8.3), we find $G(\mathrm{T}) = 2\ell^{-1}\, \mathrm{d}r\, \mathrm{T}$. Thus, $\Omega(\mathrm{T})$ is given by

$$\Omega(\mathrm{T}) = \mathrm{T}^\alpha, \qquad \alpha = 2\ell^{-1}\, \mathrm{d}r. \tag{8.16}$$

Finally, we obtain

$$-\frac{1}{2} \int_{\Sigma_r} \sqrt{-h}\, \mathcal{T}^{ab}\, \delta h_{ab} + \mathrm{d}r\, \delta \mathcal{S}_{\text{deform}} = -\frac{1}{2} \int_{\Sigma_r} \sqrt{-\tilde{h}}\, \tilde{\mathrm{T}}^{ab}\, \delta \tilde{h}_{ab}, \tag{8.17}$$

where

$$\tilde{h}_{ab} = (1 - \alpha \ln \mathrm{T}) h_{ab}, \qquad \tilde{\mathrm{T}}^{ab} = \mathrm{T}^{ab} + \alpha \left( 2 \ln \mathrm{T}\, \mathrm{T}^{ab} - \frac{\mathrm{T}}{2} h^{ab} \right). \tag{8.18}$$

From this, we conclude that in $d = 2$, radially evolving deformations indeed admit a hydrodynamic interpretation.

# 9 Summary, discussion, and outlook

The central theme of the *Freelance Holography* program (Parts I and II) is to free holography from fixed boundary and boundary conditions, employing the covariant phase space formalism (CPSF). In [34], we extended the standard gauge/gravity duality to accommodate arbitrary boundary conditions for bulk fields at the AdS asymptotic boundary. In this work, by introducing suitable boundary deformations, we pushed the AdS boundary inward, formulating a holographic framework at a finite cutoff surface (any arbitrary codimension-one timelike surface inside AdS) while allowing for arbitrary boundary conditions at the cutoff.

Previous attempts at constructing holography at a finite distance have primarily focused on imposing Dirichlet boundary conditions at the cutoff (within the $T\bar{T}$ deformation setup) [42, 55, 56]. However, it is important to note that defining holography at a finite distance with strict Dirichlet boundary conditions is not well-posed [57–64]. One of the main objectives of the current paper is to address this issue by formulating a holographic framework at a finite cutoff that allows for arbitrary boundary conditions. A detailed analysis of which boundary conditions at the cutoff lead to a well-posed theory is left for future work.

Another key objective of this program was to explore the freedoms inherent in the CPSF within the context of holography. These freedoms emerge from the application of the Poincaré lemma in both the spacetime and the field space. On a global level, they arise due to the presence of boundaries and corners. Resolving these ambiguities requires additional physical input. At the same time, the AdS/CFT dictionary establishes a correspondence between bulk and boundary theories, where the latter resides on spacetime with one lower spatial dimension. This naturally raises the question of whether holography itself provides a way to fix these freedoms. In the Freelance Holography program, we addressed these questions affirmatively both at the asymptotic AdS boundary and at a finite cutoff surface, using holography to fix the freedoms/ambiguities in both bulk and boundary descriptions.

Below, we highlight several promising directions for future research that we plan to explore:

**Null boundaries.** In this paper, we explored the radial transition from the timelike boundary of AdS spacetimes to another timelike boundary at a finite distance. An intriguing question arises: What happens if we continue moving the AdS boundary inward until it reaches a null boundary? Null boundaries are particularly interesting and important as they include black hole horizons. Moreover, null boundary arises in AdS space in Poincaré patch, the Poincaré horizon. If we can consistently deform the AdS boundary theory to reach a null boundary, we obtain a boundary theory that resides on the black hole horizon. This opens the door to investigating key problems such as the black hole membrane paradigm, black hole microstates, and the black hole information problem. Moreover, we could extend this approach further by moving the boundary inside the black hole and exploring questions related to the black hole interior.

**Interpolating between two finitely away boundaries.** In this paper, we derived the explicit form of the deformation action for an infinitesimal radial transition between two nearby boundaries. Extending this to finite-distance transitions requires integrating the bulk dynamical equations (see (6.11) for general relativity in various dimensions). However, a key point is that this integration only involves solving the radial evolution of the bulk field equations. In $d = 2$, the radial dependence of bulk fields can generally be solved exactly [65, 72, 75, 77, 93, 94]. In higher dimensions, however, exact solutions are only available for special cases. Instead of relying on exact solutions, one can approach the problem perturbatively—an approach that aligns with the construction of the solution space [83, 95].

**Other asymptotes.** In this paper, we focused on asymptotically AdS spacetimes and utilized the gauge/gravity correspondence. In recent years, $T\bar{T}$ deformations have provided new avenues for constructing holography in spacetimes with different asymptotics, particularly in the context of asymptotically dS spacetimes. The key idea is to start with AdS spacetime and use $T\bar{T}$ deformations to evolve inward until reaching a null boundary inside AdS. Since locally, one cannot distinguish between a cosmological horizon and a black hole horizon, this suggests that the deformation effectively places the theory at a cosmological horizon. From there, additional deformations—including those involving the cosmological constant—can be used to transition toward asymptotically dS spacetimes [96–102]. While this approach has been explored in the context of dS holography, the same idea is also applicable to asymptotically flat spacetimes. Developing such a flat space holography framework and comparing its implications with existing proposals [103–106] would be an interesting direction for future research.

**Radial evolution as hydrodynamic deformation.** In this paper, in addition to deformations governing radial evolution, we introduced two classes of hydrodynamic deformations—transformations that map one hydrodynamic phase space to another. This raises a natural question: Are the deformations driving radial evolution inherently hydrodynamic? We explored this question in $d = 2$ and found a positive answer. It would be interesting to extend this analysis to higher dimensions.

## Acknowledgment

We would like to thank H. Adami, B. Banihashemi, E. Shaghoulian, and M.H. Vahidinia for their useful discussions.

## A  Detailed Calculations for Section 7

In this appendix, we present the detailed calculations related to Section 7. Our primary goal is to express $\partial_r \mathcal{T}$ in terms of boundary variables.

We begin by noting that $\mathcal{T}$ consists of two parts

$$\mathcal{T} = \mathring{\mathcal{T}} + \mathcal{T}_{\text{ct}},$$

then, we compute the radial derivative of each term separately. First, for $\mathring{\mathcal{T}}$, we obtain

$$\partial_r \left( \sqrt{-h}\, \mathring{\mathcal{T}} \right) = N\sqrt{-h} \left[ (d-1)\mathring{\mathcal{O}}_{\mathcal{T}\bar{\mathcal{T}}} - \frac{d(d-1)}{\ell^2} \right] + \text{total derivative terms.} \tag{A.1}$$

This result follows from the $rr$-component of the Einstein field equations

$$\mathcal{D}_r \mathring{\mathcal{T}} = (d-1)\Box N + (d-1)N\mathring{\mathcal{T}}_{ab}\mathring{\mathcal{T}}^{ab} - \frac{d-2}{d-1}N\mathring{\mathcal{T}}^2 - \frac{d(d-1)}{\ell^2}N. \tag{A.2}$$

Next, we compute $\partial_r \left( \sqrt{-h}\, \mathcal{T}_{\text{ct}} \right)$. The counterterm $\mathcal{T}_{\text{ct}}$ depends on the spacetime dimension, as given in Eq. (6.6)

$$d = 2: \quad \kappa\,\mathcal{T}_{\text{ct}} = \frac{2}{\ell}, \qquad d = 3: \quad \kappa\,\mathcal{T}_{\text{ct}} = \frac{6}{\ell} + \frac{\ell}{2}\hat{R}, \qquad d = 4: \quad \kappa\,\mathcal{T}_{\text{ct}} = \frac{12}{\ell} + \frac{\ell}{2}\hat{R}. \tag{A.3}$$

Now, we compute the radial derivative for each case separately. Straightforward calculations yield:

$$d = 2: \quad \partial_r \left( \sqrt{-h}\, \mathcal{T}_{\text{ct}} \right) = -2\ell^{-1} N \sqrt{-h}\, \mathring{\mathcal{T}} + \text{total derivative terms},$$

$$d = 3: \quad \partial_r \left( \sqrt{-h}\, \mathcal{T}_{\text{ct}} \right) = -N \sqrt{-h} \left[ \ell \hat{R}_{ab} \mathring{\mathcal{T}}^{ab} + \frac{12 - \ell^2 \hat{R}}{4\ell} \mathring{\mathcal{T}} \right] + \text{total derivative terms},$$

$$d = 4: \quad \partial_r \left( \sqrt{-h}\, \mathcal{T}_{\text{ct}} \right) = -N \sqrt{-h} \left[ \ell \hat{R}_{ab} \mathring{\mathcal{T}}^{ab} + \frac{24 - \ell^2 \hat{R}}{6\ell} \mathring{\mathcal{T}} \right] + \text{total derivative terms}. \tag{A.4}$$

Where, we have used the condition $\hat{\nabla}_a \mathring{\mathcal{T}}^{ab} = 0$. Combining Eqs. (A.1) and (A.3), we obtain:

$$d = 2: \quad \int_{\Sigma_r} \partial_r(\sqrt{-h}\,\mathcal{T}) = \int_{\Sigma_r} N\sqrt{-h} \left[ \mathring{\mathcal{O}}_{\mathcal{T}\bar{\mathcal{T}}} - \frac{2\mathring{\mathcal{T}}}{\ell} - \frac{2}{\ell^2} \right],$$

$$d = 3: \quad \int_{\Sigma_r} \partial_r(\sqrt{-h}\,\mathcal{T}) = \int_{\Sigma_r} N\sqrt{-h} \left[ 2\mathring{\mathcal{O}}_{\mathcal{T}\bar{\mathcal{T}}} - \frac{6}{\ell^2} - \ell \hat{R}_{ab} \mathring{\mathcal{T}}^{ab} - \frac{12 - \ell^2 \hat{R}}{4\ell} \mathring{\mathcal{T}} \right], \tag{A.5}$$

$$d = 4: \quad \int_{\Sigma_r} \partial_r(\sqrt{-h}\,\mathcal{T}) = \int_{\Sigma_r} N\sqrt{-h} \left[ 3\mathring{\mathcal{O}}_{\mathcal{T}\bar{\mathcal{T}}} - \frac{12}{\ell^2} - \ell \hat{R}_{ab} \mathring{\mathcal{T}}^{ab} - \frac{24 - \ell^2 \hat{R}}{6\ell} \mathring{\mathcal{T}} \right].$$

Finally, rewriting these expressions in terms of the rBY-EMT, we obtain the final result:

$$d = 2: \quad \int_{\Sigma_r} \partial_r(\sqrt{-h}\,\mathcal{T}) = \int_{\Sigma_r} N\sqrt{-h}\, \mathcal{O}_{\mathcal{T}\bar{\mathcal{T}}},$$

$$d = 3: \quad \int_{\Sigma_r} \partial_r(\sqrt{-h}\,\mathcal{T}) = \int_{\Sigma_r} N\sqrt{-h} \left[ 2\mathcal{O}_{\mathcal{T}\bar{\mathcal{T}}} + \frac{\mathcal{T}}{\ell} - \frac{3}{4}\ell \hat{R}\mathcal{T} + \ell^2 \hat{R}_{ab}\hat{R}^{ab} - \frac{3}{8}\ell^2 \hat{R}^2 + 3\ell \hat{R}_{ab}\mathcal{T}^{ab} \right], \tag{A.6}$$

$$d = 4: \quad \int_{\Sigma_r} \partial_r(\sqrt{-h}\,\mathcal{T}) = \int_{\Sigma_r} N\sqrt{-h} \left[ 3\mathcal{O}_{\mathcal{T}\bar{\mathcal{T}}} + \frac{2\mathcal{T}}{\ell} + \frac{\ell}{3}\hat{R}\mathcal{T} + \frac{\ell^2}{4}\hat{R}_{ab}\hat{R}^{ab} - \frac{\ell^2}{12}\hat{R}^2 + 2\ell \hat{R}_{ab}\mathcal{T}^{ab} \right].$$

We conclude this section by noting that the constraint equations (6.7a), expressed in terms of the rBY energy-momentum tensor, are given by

$$d = 3: \quad \kappa^2\, \mathcal{O}_{\mathcal{T}\bar{\mathcal{T}}} + 2\kappa \frac{\mathcal{T}}{\ell} + 2\kappa\, \ell \hat{R}_{ab}\, \mathcal{T}^{ab} - \frac{1}{2}\kappa\, \ell \hat{R}\mathcal{T} + \ell^2 \left( \hat{R}_{ab}\,\hat{R}^{ab} - \frac{3}{8}\hat{R}^2 \right) = 0,$$

$$d = 4: \quad \kappa^2\, \mathcal{O}_{\mathcal{T}\bar{\mathcal{T}}} + 2\kappa \frac{\mathcal{T}}{\ell} + \kappa\, \ell \hat{R}_{ab}\, \mathcal{T}^{ab} - \frac{1}{6}\kappa\, \ell \hat{R}\mathcal{T} + \frac{\ell^2}{12} \left( 3\hat{R}_{ab}\,\hat{R}^{ab} - \hat{R}^2 \right) = 0. \tag{A.7}$$

It is interesting to note that the last terms in parentheses in the equations above are related to higher curvature terms in New Massive Gravity (NMG) [107] and critical gravity actions [108], respectively. We will explore this observation in more detail in future work.

# B    Radial transformations

This section shows how different components of the covariant spin 0, 1, and 2 fields in the radial ADM decompositions transform under diffeomorphisms. Take $\xi = \xi^\mu \partial_\mu = \xi\, \partial_r + \bar{\xi}^a \partial_a$ to be a generic diffeomorphism, then a scalar, a one-form, and a metric transform respectively as:

**Scalar field.**    For a scalar field, $\Phi$, we have

$$\delta_\xi \Phi = \mathcal{L}_\xi \Phi = \xi\, \partial_r \Phi + \mathcal{L}_{\bar{\xi}} \Phi. \tag{B.1}$$

**One-form field.** Also, for a one-form field, $A_\mu \, \mathrm{d}x^\mu = \phi \, \mathrm{d}r + A_a \, \mathrm{d}x^a$, we find

$$\delta_\xi \phi = \partial_r(\phi \, \xi) + \mathcal{L}_{\bar\xi} \phi + A_a \partial_r \bar\xi^a \,, \tag{B.2a}$$

$$\delta_\xi A_a = \xi \, \partial_r A_a + \mathcal{L}_{\bar\xi} A_a + \phi \, \partial_a \xi \,. \tag{B.2b}$$

**Metric.** The components of the metric (2.1) transform as follows

$$\delta_\xi N = \partial_r(N \, \xi) + \mathcal{L}_{\bar\xi} N - N \, U^a \partial_a \xi \,, \tag{B.3a}$$

$$\delta_\xi U^a = \partial_r(\bar\xi^a + \xi \, U^a) + N^2 h^{ab} \partial_b \xi + \mathcal{L}_{\bar\xi} U^a - U^a \mathcal{L}_U \xi \,, \tag{B.3b}$$

$$\delta_\xi h_{ab} = \mathcal{L}_{\bar\xi} h_{ab} + \xi \, \partial_r h_{ab} + 2 U_{(a} \partial_{b)} \xi \,. \tag{B.3c}$$

**Radial transformation.** For the case $\xi = \partial_r$, the variations are given by

$$\delta_{\partial_r} \Phi = \partial_r \Phi \,, \qquad \delta_{\partial_r} A_\mu = \partial_r A_\mu \,, \qquad \delta_{\partial_r} g_{\mu\nu} = \partial_r g_{\mu\nu} \,. \tag{B.4}$$

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
