# Peer review of "Freelance Holography, Part II: Moving Boundary in Gauge/Gravity Correspondence"

_SciPost Physics_

## Round 2 · Referee Report · Anonymous (Referee 1) · 2025-6-19

Strengths

  • The problem being considered in this paper is ambitious and very relevant.
  • The explanations provided are clear.
  • The proposal is illustrated with several examples.

Weaknesses

  • The proposal seems to lack an independent field theory definition.
  • The proposal is insufficiently grounded at the quantum level as the effects of irrelevant operators in the RG flow are not considered.
  • There are certain claims that should be clearified - see the report.

Report

This paper proposes an extension of the AdS/CFT correspondence that applies to gravitational systems bounded by a generic timelike codimension-one surface, with arbitrary boundary conditions. The methodology is that of the covariant phase space formulatism in the bulk. Various examples of radial evolution in AdS GR + matter theories are worked out, and their hydrodynamic interpretation is discussed.

Although the problem considered in this paper is a very relevant one that falls within the scope of SciPost Physics, there remain considerable open questions on the consistency and scope of this proposal, outlined below. There are some further concrete issues listed in the “requested changes” paragraph.

1- Field theory flow operator

It is claimed in the abstract that the evolution “from the [AdS] boundary to another boundary inside AdS [...] is encoded in deformations of the holographic boundary theory.” However, there does not seem to be a purely field theoretic definition of the deformation. For example, the deformation action $\mathcal S_\text{deform}$ in (3.25) is given in terms of the on-shell bulk Lagrangian, which is not a boundary quantity. In particular, it requires the knowledge of how the radial derivative of the bulk field is related to its conjugate momentum. See, for example, the term $\partial_r J^i$ in (6.9) which seems mysterious from the field theory point of view. Also in sections 6.3 and 6.4, knowledge of the bulk equations is required to define what the flow operator is.

This stands in contrast with the $T \bar T$ operator, which is defined purely from field theory ingredients. (Here one should note, however, that this operation can only be interpreted as moving the boundary in the theory dual to pure gravity.)

To make this point a bit more concrete, one can ask this question in the specific example of the duality between type IIB string theory in AdS$_5 \times S^5$ and $\mathcal N = 4$ SYM: what gauge theory operator does your prescription predict one should add to the SYM action to let it flow into the bulk?

As a related remark, it is not clear what the content of equation (3.35) is, since the "QFT" on the left-hand side, as far as I can see, has not been given a precise definition.

2- Classical well-posedness of Lorentzian boundary conditions in the bulk

It is mentioned that finite radial cutoff and Dirichlet boundary conditions are ill-defined in $D > 3$ spacetime dimensions. However, this statement does not seem to be addressed in the rest of the paper. Even though the conclusion section mentions that one of the objectives was to address this issue, it is not clear to me how the current proposal solves this problem, or whether it is unresolved and left as an open issue.

3- Relevance of double-trace deformations & well-definedness of CFT

Already before flowing into the bulk, the authors should clarify to what extent their proposal for altered boundary conditions using CPSF freedoms is rigorous. For example, a generic boundary term $W_\text{bulk}$ will contain operators that are irrelevant in the CFT. To take this proposal beyond the classical level, one should therefore restrict to the small set of multi-trace operators which are relevant and trigger a flow to an IR CFT. (One example that might be worth investigating in more detail is the free/critical $O(N)$ model in 3d.) Alternatively, one would have to argue, as it is done for the special case of the $T \bar{T}$ operator in two dimensions, that the irrelevant deformation is somehow tractable. (For example, the somewhat related prescription by Compère and Marolf presented some arguments that their construction is UV complete.)

Alternatively, the authors should make it clear that one should only expect the dual field theory to be an effective description, as was done for example for the higher-dimensional $T \bar{T}$ results by Taylor and by Hartman, Kruthoff, Shaghoulian and Tajdini.

Requested changes

  • The authors should address the issues raised in the main report.
  • It is claimed in the introduction that the bulk $Y$ freedom is uniquely determined in terms of the boundary symplectic potential. However, near (4.12) of part I of this paper series, this is merely called a “convenient choice” rather than a unique determination. The status of this result should be clarified.
  • The authors should clarify the discussion under equation (2.10). Is the equation $d \delta \Theta_D = 0$ supposed to hold off-shell, as there is no circle above the equal sign? If so, also $d \delta \Theta = 0$ off shell. In any case, it does not seem like this condition specifies $\Theta_D$ uniquely. Also the fact that it is compatible with Dirichlet boundary conditions doesn't uniquely fix it, because any function of the source on the boundary can still be added to the boundary action.
  • I object to the term "gauge/gravity correspondence" being used to specifically indicate the AdS/CFT duality in the 't Hooft limit, as is done on page 6.

Recommendation

Ask for major revision

  • validity: -
  • significance: top
  • originality: good
  • clarity: good
  • formatting: perfect
  • grammar: perfect

Author:  Shahin Sheikh-Jabbari  on 2025-07-02  [id 5613]

(in reply to Report 1 on 2025-06-19)

Please see the attached file for the detailed reply to the comments by both referees.

Attachment:

Freelance-II-SciPost-Reply_UKLCiCT.pdf

---

## Round 2 · Referee Report · Anonymous (Referee 2) · 2025-6-24

Strengths

  1. The paper gives a clear and self contained review of the background in holography at finite radius.
  2. The paper sets out the new results in sections 4 and 5, before giving specific examples illustrating these in the following sections.

Weaknesses

  1. This work follows as a direct generalization of earlier literature on holography at finite radius, and does not add significant new conceptual results.
  2. This paper does not address the subtleties around whether the operators defined at finite radius are actually local operators.

Report

This work continues a programme of research exploring holography at finite radius. The main new content of this work relates to generalizing the boundary condition at finite radius from purely Dirichlet, and the consideration of interpolating boundary conditions. Both are straightforward generalisations of existing literature.

The paper does not address various known technical subtleties. One subtlety is that the formula (3.11), and subsequent related formulae, do not apply when the operator dimension is $d/2 + n$ where $n$ is an integer. In such contexts the relationship between operator source and expectation value is slightly different, due to the conformal anomalies. Note that this situation is highly generic: the most interesting operators in even spacetime dimensional theories indeed do have integral dimensions. This technical subtlety could be acknowledged by a suitable comment and reference.

A second subtlety relates to the assumption implicit in (3.18) about the identification of operator source at finite radius. This is not a new assumption from these authors, but there is a longstanding question on how exactly the holographic radius relates to the renormalisation scale and it is not clear that the operator identified in (3.18) would be local. Again this issue could be acknowledged by comments and references.

Requested changes

I would recommend that this paper is published in Scipost Core, with the minor clarifications mentioned above addressed through suitable comments and references.

Recommendation

Accept in alternative Journal (see Report)

  • validity: good
  • significance: good
  • originality: good
  • clarity: high
  • formatting: excellent
  • grammar: excellent

Author:  Shahin Sheikh-Jabbari  on 2025-07-02  [id 5612]

(in reply to Report 2 on 2025-06-24)
Category:
correction

Please see the attached file for a detailed reply.

Attachment:

Freelance-II-SciPost-Reply.pdf

---

## Editorial Decision

resubmitted